# Neural Evolution Strategy for Black-box Pareto Set Learning

**Chengyu Lu**[1,2], **Zhenhua Li**[3,4,*], **Xi Lin**[1,2], **Ji Cheng**[1,2], **Qingfu Zhang**[1,2,*]

[1] City University of Hong Kong, [2] City University of Hong Kong Shenzhen Research Institute
[3] Nanjing University of Aeronautics and Astronautics
[4] MIIT Key Laboratory of Pattern Analysis and Machine Intelligence
{chengyulu3-c, xi.lin, J.Cheng}@my.cityu.edu.hk, zhenhua.li@nuaa.edu.cn
qingfu.zhang@cityu.edu.hk

## Abstract

Multi-objective optimization problems (MOPs) are prevalent in numerous real-world applications. Recently, Pareto Set Learning (PSL) has emerged as a powerful paradigm for solving MOPs. PSL can produce a neural network for modeling the set of all Pareto optimal solutions. However, applying PSL to black-box objectives, particularly those exhibiting non-separability, high dimensionality, and/or other complex properties, remains very challenging. To address this issue, we propose leveraging evolution strategies (ESs), a class of specialized black-box optimization algorithms, within the PSL paradigm. Traditional ESs capture the complex dimensional dependencies less efficiently, which can significantly hinder their performance in PSL. To tackle this issue, we suggest encapsulating the dependencies within a neural network, which is then trained using a novel gradient estimation method. The proposed method, termed Neural-ES, is evaluated using a bespoke benchmark suite for black-box PSL. Experimental comparisons with other methods demonstrate the efficiency of Neural-ES, underscoring its ability to learn the Pareto sets of challenging black-box MOPs.

## 1 Introduction

Many real-life applications need to consider multiple conflicting objectives simultaneously [1–3]. Since these objectives can rarely be optimized by a single solution simultaneously, a commonly used approach is to find the *Pareto set* (PS) [4], which contains all solutions with different optimal trade-offs among the objectives. Traditional methods [5–8] approximate the PS using only a finite number of solutions, risking not aligning with users' trade-off preferences determined after optimization.

Recently, a novel paradigm called *Pareto Set Learning* (PSL) has attracted increasing attention [9–15]. PSL formulates a multi-objective optimization problem (MOP) [4] as a set learning task by decomposing the problem into an *infinite* number of single-objective subproblems and resolving the optimal solutions for all of them. In this process, a *set model*, typically a neural network, learns the mapping from an input preference to its corresponding Pareto optimal solution. Consequently, the Pareto set is recovered in both the objective and solution spaces. Unlike traditional multi-objective methods, which only accommodate a limited number of predefined user preferences [16, 5], a trained PSL model can provide a tailored optimal solution for any valid decision-maker preference trade-off without requiring re-optimization from scratch.

Despite the rapid development of PSL, it remains largely unclear how to apply this paradigm to a black-box MOP where both the analytic expression and derivatives of the objectives are unknown. On

---

[*]Correspondence to: Qingfu Zhang and Zhenhua Li

39th Conference on Neural Information Processing Systems (NeurIPS 2025).

one hand, black-box objectives are prevalent in practice [17, 18, 7], making it imperative to develop novel approaches for effectively tackling black-box MOPs. On the other hand, when exhibiting common properties such as non-separability and high dimensionality, black-box objectives can become highly challenging, rendering their resolution an open question. Moreover, since PSL involves solving numerous subproblems that may themselves be black boxes, jointly optimizing them poses a significantly greater challenge than solving a single black-box optimization problem. Lin *et al.* [9] proposed integrating a Gaussian process model into PSL for solving expensive MOPs. However, Gaussian process models scale poorly with problem dimensions and increased evaluation budgets, which may be less suited for more general black-box MOPs.

Evolution Strategy (ES) has remained one of the most successful black-box optimization algorithms for decades [19–21]. An ES algorithm utilizes a distribution model, typically a multivariate Gaussian distribution, to search in complex fitness landscapes. Key elements that govern a competitive ES algorithm include an adaptive covariance matrix, the resemblance to gradient search, and sophisticated adaptation strategies. Compared to other zeroth-order optimization methods that sample solutions from isotropic distributions [22, 23], ESs excel at dealing with variables' complex *dimensional dependencies*, a prevalent feature as well as a significant challenge in real-world optimization problems [24–26]. It is natural to integrate PSL with ES for black-box multi-objective optimization. On one hand, PSL decomposes an MOP into single-objective subproblems, aligning perfectly with the strengths of ES. On the other hand, ES's search efficiency paves the way for dealing with challenging black-box objectives. However, traditional ES algorithms experience high computational complexity [27, 28] and rely heavily on adaptation strategies [29–31]. These limitations are particularly crucial and evident within PSL, as they can lead to failures in decoupling dimensional dependencies and significantly hinder the performance of ES.

To enhance the PSL paradigm for solving black-box MOPs, we propose a novel method called Neural-ES, which acts as the back-end of the set model. To effectively handle dimensional dependencies, we propose rethinking the way ES algorithms decouple such dependencies. Rather than leveraging a covariance matrix, Neural-ES encapsulates the dependencies within a neural network. To be more specific, we partition the high-dimensional design variable of a black-box objective into lower-dimensional segments. Each segment follows a distinct multivariate Gaussian distribution. The network, which also acts as the set model, parameterizes the distribution conditioned on all previously sampled segments. To train the network, we estimate the gradient in a novel way closely related to the natural gradient [32, 33]. Neural-ES is well-suited for black-box tasks involving neural networks, with PSL serving as a typical example. We summarize our contributions in threefold:

- We propose Neural-ES, a novel algorithm that utilizes a neural network to capture complex dimensional dependencies effectively, which enhances the versatility of the ES repertoire.

- We establish a novel PSL formulation for black-box MOPs, which enables the use of efficient derivative-free optimization algorithms to address the problem efficiently.

- We conduct extensive experiments using a novel benchmark suite designed for evaluating black-box PSL problems. Results demonstrate that Neural-ES effectively addresses non-separable MOPs. To the best of our knowledge, we are the first to scale the PSL paradigm to over 1,000 dimensions in black-box scenarios.

## 2 Preliminaries & related works

### 2.1 Multi-objective optimization

A continuous MOP (MOP) [4] is defined as:

$$\underset{\boldsymbol{x} \in \mathcal{X}}{\text{minimize}} \, \boldsymbol{F}(\boldsymbol{x}) = (f_1(\boldsymbol{x}), ..., f_M(\boldsymbol{x}))^T, \tag{1}$$

where $\boldsymbol{F} : \mathcal{X} \to \mathbb{R}^M$ consists of $M$ objectives $f_i : \mathcal{X} \to \mathbb{R}$ for all $i = 1, ..., M$. In many cases, the objectives conflict with each other, and a single solution cannot optimizes all of them simultaneously. An alternative is to pursue the *Pareto set*, which consists of optimal solutions under all possible trade-offs among the objectives. Collectively, these solutions exclude any opportunity to improve one objective without compromising at least one other. We provide formal definitions of the fundamentals of MOPs in the following:

**Definition 1** (Pareto dominance). Let $\boldsymbol{x}, \boldsymbol{x}^{'} \in \mathcal{X}$ be two solutions to (1), $\boldsymbol{x}^{'}$ is said to dominate $\boldsymbol{x}$, denoted by $\boldsymbol{x}^{'} \prec \boldsymbol{x}$, if $f_i(\boldsymbol{x}^{'}) \leq f_i(\boldsymbol{x}), \forall i = 1, ..., M$, and $f_j(\boldsymbol{x}^{'}) < f_j(\boldsymbol{x}), \exists j = 1, ..., M$.

**Definition 2** (Pareto optimality). A solution $\boldsymbol{x} \in \mathcal{X}$ is Pareto optimal if $\boldsymbol{x}^{'} \nprec \boldsymbol{x}, \forall \boldsymbol{x}^{'} \in \mathcal{X}$.

**Definition 3** (Pareto set & front). The set of all the Pareto optimal solutions is called the Pareto set, denoted by $PS = \{\boldsymbol{x} \in \mathcal{X} | \nexists \boldsymbol{x}^{'}, \boldsymbol{x}^{'} \prec \boldsymbol{x}\}$; the image of $PS$ in the objective space is called the Pareto front, denoted by $PF = \{\boldsymbol{F}(\boldsymbol{x}) | \boldsymbol{x} \in PS\}$.

**Multi-objective evolutionary algorithms (MOEAs)**   MOEAs have developed as one of the key methodologies for solving MOPs [16, 5, 6]. Compared to other methodologies [34, 7], MOEAs approximate the PS by a population of diverse non-dominated solutions, without collapsing into a single point. Several highly efficient MOEA variants have been developed [6, 35, 36]. However, MOEAs encounter longstanding issues, including inefficient Pareto ranking, misalignment with decision-makers' preferences, poor scalability in relation to population size, and ineffective information sharing among subproblems.

## 2.2   Pareto set learning

PSL formulates an MOP as follows. First, given a preference vector $\boldsymbol{\alpha} = (\alpha_1, ..., \alpha_M)^T \in \Delta_M$ where $\Delta_M = \{\boldsymbol{\alpha} \in \mathbb{R}_+^M | \mathbf{1}^T \boldsymbol{\alpha} = 1\}$ is the $(M-1)$-dimensional simplex, and an aggregation function $g : \mathbb{R}^M \to \mathbb{R}$ conditioned on $\boldsymbol{\alpha}$, (1) can be aggregated into a single-objective subproblem. Under some mild conditions, solving every possible subproblem conditioned on all possible $\boldsymbol{\alpha}$ recovers the entire Pareto set. Then, an MOP in (1) can be formulated as:

$$\underset{\boldsymbol{\theta} \in \Theta}{\text{minimize}} \, \mathbb{E}_{\boldsymbol{\alpha} \sim \mathcal{U}(\Delta_M)}[g(\boldsymbol{x}; \boldsymbol{\alpha})], \text{ where } \boldsymbol{x} = \boldsymbol{\varphi_\theta}(\boldsymbol{\alpha}), \tag{2}$$

where $\Theta$ is the parameter space, $\mathcal{U}(\Delta_M)$ denotes a probability distribution (in this work, it is set as a uniform distribution) over $\Delta_M$, and $\boldsymbol{\varphi_\theta} : \Delta_M \to \mathcal{X}$ is called a *set model* parameterized by $\boldsymbol{\theta}$. In the ideal case, $\boldsymbol{\varphi_\theta}(\boldsymbol{\alpha}) = \text{argmin}_{\boldsymbol{x} \in \mathcal{X}} \, g(\boldsymbol{x}; \boldsymbol{\alpha}), \forall \boldsymbol{\alpha} \in \Delta_M$, which means $\boldsymbol{\varphi_\theta}$ maps each preference to its corresponding Pareto optimal solution that minimizes $g(\cdot; \boldsymbol{\alpha})$.

**Recent developments**   Since its inception, PSL [9] has been rapidly applied to various problem types, including expensive [13, 11], combinatorial [37], and smooth MOPs [38]. However, these methods face challenges with general black-box MOPs, particularly when the objectives are high-dimensional and exhibit complex dimensional dependencies. Lin *et al.* [15] contributed a primary work called evolutionary-PSL (EPSL), where a finite difference approach is employed to train the set model. However, due to the absence of a comprehensive distribution model, most existing derivative-free PSL algorithms have been validated only on low-dimensional toy problems, leaving their performance on challenging problems uncertain.

## 2.3   Evolution strategy

ES is a prominent class of derivative-free optimization algorithms. Popular algorithms include natural evolution strategy (NES) [28, 20], the state-of-the-art covariance matrix adaptation (CMA-ES) [19, 27], and the also competitive but more time-efficient matrix adaptation evolution strategy (MA-ES) [39]. An ES algorithm first formulates a general black-box optimization problem in question as:

$$\underset{\boldsymbol{x} \in \mathcal{X}}{\text{minimize}} \, f(\boldsymbol{x}) \Rightarrow \underset{\boldsymbol{\varphi} \in \Phi}{\text{maximize}} -\mathbb{E}_{\pi(\boldsymbol{x}; \boldsymbol{\varphi})}[f(\boldsymbol{x})], \tag{3}$$

where $\mathcal{X} \subseteq \mathbb{R}^n$ is the search space, $\pi(\boldsymbol{x}; \boldsymbol{\varphi})$ is the probability density function, *p.d.f.*, of a smooth distribution $\boldsymbol{x} \sim \pi(\boldsymbol{\varphi})$ parameterized by $\boldsymbol{\varphi}$, and $\Phi$ is the parameter space of $\boldsymbol{\varphi}$. At each $x \in \mathcal{X}$, only its $f$ function value $f(x)$ is available. Most commonly, $\pi(\boldsymbol{\varphi}) = \mathcal{N}(\boldsymbol{m}, \boldsymbol{C})$ is a multivariate Gaussian distribution where $\boldsymbol{\varphi}$ consists of the distribution mean $\boldsymbol{m} \in \mathbb{R}^n$ and the covariance matrix $\boldsymbol{C} \in \mathbb{S}_{++}^n$ (where $\mathbb{S}_{++}^n$ denotes the space of $n$-dimensional symmetric and positive definite matrices). In brief, an ES optimizes $f$ by 1) sampling random samples from $\pi(\boldsymbol{\varphi})$ and 2) effectively updating $\boldsymbol{\varphi}$.

**Covariance matrix and dimensional dependencies**   It is widely acknowledged that the covariance matrix plays a vital role in the performance of an ES algorithm. Hansen *et al.* [24] systematically

showed that CMA-ES, aided by an adaptive covariance matrix, effectively decouples the interacting problem dimensions. Beyer [26] showed that the covariance matrix decouples the dimensional dependencies by effectively converging to the inverse of a convex objective's Hessian. Both Akimoto *et al.* [40] and Glasmarchers *et al.* [28] demonstrated that the rank-$\mu$ of CMA-ES amounts to the natural gradient ascent method. The ability to decouple dimensional dependencies positions NES and CMA-ES as second-order quasi-Newton methods rather than merely being zeroth-order finite-difference methods.

**Adaptation strategies**  In addition to natural gradient, refined adaptation strategies also contribute to the competitiveness of ESs, as exemplified by cumulative step-length control and evolution paths [19, 41]. An evolution path accumulates the movement trajectories of the distribution over time, thereby exploiting a promising search direction. Evidence suggests that ESs leveraging both natural gradient and adaptation strategies significantly outperform those that rely solely on natural gradient [28, 29].

## 3  Black-box Pareto set learning

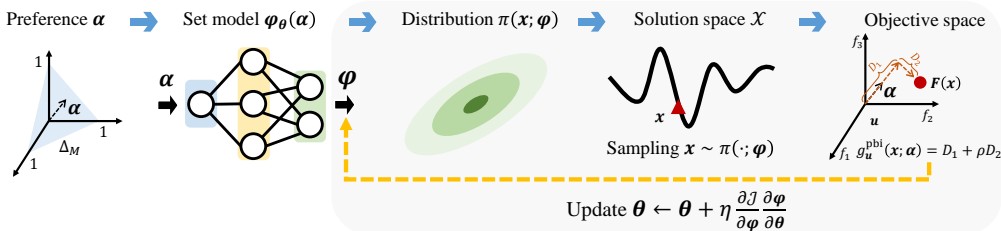

Figure 1: Black-box PSL formulation. The blue arrows indicate the forward process, and the yellow arrow indicates the gradient back-propagation; the shadowed area is where an ES participates and is referred to as the *back-end* of the set model.

### 3.1  Problem formulation

A black-box MOP refers to those $\boldsymbol{F}$ in (1) that provide no information beyond the objective values of a solution. In this case, we formulate (2) through the *reparameterization* trick:

$$\operatorname*{maximize}_{\boldsymbol{\theta}\in\Theta} \mathcal{J}(\boldsymbol{\theta}) = \mathbb{E}_{\boldsymbol{\alpha}\sim\mathcal{U}(\Delta_M)}\mathbb{E}_{\pi(\boldsymbol{x};\boldsymbol{\varphi})}[-g(\boldsymbol{x};\boldsymbol{\alpha})], \text{ where } \boldsymbol{\varphi} = \boldsymbol{\varphi}_{\boldsymbol{\theta}}(\boldsymbol{\alpha}). \tag{4}$$

The above formulation (4) distinguishes itself from (2) in the nested expectation. Instead of outputting $\boldsymbol{x}$ directly, the set model $\boldsymbol{\varphi}_{\boldsymbol{\theta}}(\boldsymbol{\alpha})$ first maps a preference vector $\boldsymbol{\alpha}$ to a distribution $\pi(\boldsymbol{\varphi})$ parameterized by $\boldsymbol{\varphi} = \boldsymbol{\varphi}_{\boldsymbol{\theta}}(\boldsymbol{\alpha})$, and then samples solutions from the distribution. To train the set model, an ES is employed to sample solutions and estimate gradients. When $\boldsymbol{\theta}$ is perfectly learned, regarding an arbitrary $\boldsymbol{\alpha}$, a random $\boldsymbol{x} \sim \pi(\boldsymbol{\varphi}_{\boldsymbol{\theta}}(\boldsymbol{\alpha}))$ converges in probability to the minimum of $g(\cdot;\boldsymbol{\alpha})$.

To further illustrate this process, we outline the paradigm of black-box PSL in Fig. 1. First, a random preference vector $\boldsymbol{\alpha} \sim \mathcal{U}(\Delta_M)$ is sampled and inputted into the set model $\boldsymbol{\varphi}_{\boldsymbol{\theta}}$. The set model subsequently outputs a parameter vector $\boldsymbol{\varphi} \in \Phi$, which parameterizes a distribution $\pi(\boldsymbol{x};\boldsymbol{\varphi})$. A random solution $\boldsymbol{x} \sim \pi(\boldsymbol{\varphi})$ is then sampled from this distribution and evaluated on the specific $g(\cdot;\boldsymbol{\alpha})$. Here, we use the PBI aggregation function $g_{\boldsymbol{u}}^{\mathrm{pbi}}$ [8] as an example. When we have sampled enough $\boldsymbol{\alpha}$s, and for each $\boldsymbol{\alpha}$ there have been several samples $\boldsymbol{x}$s, we can estimate the gradient and back-propagate it to $\boldsymbol{\theta}$ through the chain rule.

### 3.2  Key challenges

The formulation (4) introduces several challenges for ES. First, the high computational complexity of the covariance matrix significantly impedes an ES's performance. It takes $\Theta(n^2)$ space complexity of the set model to generate a covariance matrix, and $\Theta(n^4)$ time complexity or even more to compute the gradient, which will soon become prohibitive as $n$ increases. Even worse, since each distribution $\boldsymbol{\varphi}$ corresponds to a different $\boldsymbol{\alpha}$, as can be seen from Fig. 1, the large number of decomposed subproblems

in PSL necessitates an equally large number of covariance matrices, which drastically increases the learning burden of the set model.

Second, it is difficult, if not impossible, to apply sophisticated adaptation strategies in PSL problems. These strategies are known to accelerate convergence by counteracting the slow-changing rate suppressed by the Kullback-Leibler (KL)-divergence [26]. Despite some recent findings [42], the connection between these strategies and gradient search remains unclear, which limits their use in training the set model. Furthermore, as there are numerous decomposed subproblems in a PSL problem, it is infeasible to track a different evolution path for each subproblem.

The above characteristics of a black-box PSL problem make the direct integration of ESs rarely yield competitive performance, posing new challenges for ESs to learn in a typical adaptation-free scenario.

## 4 Neural evolution strategy

### 4.1 Basic idea

Consider a general single-objective problem: $\text{maximize}_{\boldsymbol{\theta}}\ \bar{\mathcal{J}}_{\boldsymbol{\alpha}}(\boldsymbol{\theta}) = \mathbb{E}_{\pi(\boldsymbol{x};\boldsymbol{\varphi})}[f(\boldsymbol{x})]$, where $\boldsymbol{\varphi} = \boldsymbol{\varphi}_{\boldsymbol{\theta}}(\boldsymbol{\alpha})$, $\boldsymbol{\alpha}$ is a problem specific parameter, and $\boldsymbol{x} \in \mathbb{R}^n$. Assuming $n = p \times d$ where $p$ and $d$ are positive integers, $\boldsymbol{x}$ can be partitioned into $p$ segments each of length $d$, as follows:

$$\boldsymbol{x} = \big(\underbrace{\mathrm{x}_1, ..., \mathrm{x}_d}_{\boldsymbol{x}_1^T}, ..., \underbrace{\mathrm{x}_{(i-1)d+1}, ..., \mathrm{x}_{id}}_{\boldsymbol{x}_i^T}, ..., \underbrace{\mathrm{x}_{n-d+1}, ..., \mathrm{x}_n}_{\boldsymbol{x}_p^T}\big)^T \qquad (5)$$

By conditional probability, $\pi(\boldsymbol{x};\boldsymbol{\varphi}) = \pi(\boldsymbol{x};\boldsymbol{\varphi}_{\boldsymbol{\theta}}(\boldsymbol{\alpha}))$ can be decomposed into the product of a chain of *p.d.f.*s each concerning a different $\boldsymbol{x}_i$, and conditioned on all the previous $\boldsymbol{x}_0, \boldsymbol{x}_1, ..., \boldsymbol{x}_{i-1}$:

$$\pi(\boldsymbol{x};\boldsymbol{\varphi}_{\boldsymbol{\theta}}(\boldsymbol{\alpha})) = \prod_{i=1}^{p} \mathcal{N}_{\boldsymbol{\theta}}(\boldsymbol{x}_i | \underbrace{\boldsymbol{x}_0, \boldsymbol{x}_1, ..., \boldsymbol{x}_{i-1}}_{\boldsymbol{x}_{0:i-1}}) = \prod_{i=1}^{p} \mathcal{N}\left(\boldsymbol{x}_i; \boldsymbol{\varphi}_{\boldsymbol{\theta}}(\boldsymbol{x}_{0:i-1})\right). \qquad (6)$$

With a slight abuse of symbols, we redefine $\boldsymbol{\varphi}_{\boldsymbol{\theta}} : \bigcup_{i=1}^{n} \mathbb{R}^i \to \mathbb{R}^d \times \mathbb{S}_{++}^d$ hereafter, which means $\boldsymbol{\varphi}_{\boldsymbol{\theta}}$ maps variable-length inputs to the parameters of a $d$-dimensional Gaussian. More specifically, an input $\boldsymbol{x}_{0:i-1}$ consists of all the segments $\boldsymbol{x}_0, \boldsymbol{x}_1..., \boldsymbol{x}_{i-1}$ preceding $\boldsymbol{x}_i$, and the output $\boldsymbol{\varphi}_{\boldsymbol{\theta}}(\boldsymbol{x}_{0:i-1})$ consists of the mean $\boldsymbol{m}_{\boldsymbol{\theta}}(\boldsymbol{x}_{0:i-1}) \in \mathbb{R}^d$ and the covariance matrix $\boldsymbol{C}_{\boldsymbol{\theta}}(\boldsymbol{x}_{0:i-1}) \in \mathbb{S}_{++}^d$. $\boldsymbol{x}_0$ is just an alias of $\boldsymbol{\alpha}$, for notation consistency. (6) directly models the dependency of $\boldsymbol{x}_i$ on $\boldsymbol{x}_{0:i-1}$ as the mapping $\boldsymbol{\varphi}_{\boldsymbol{\theta}}$. In practice, $\boldsymbol{\varphi}_{\boldsymbol{\theta}}$ is a neural network. Fig. 2 illustrates the idea.

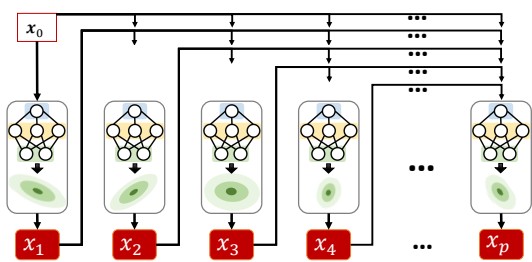

Figure 2: An illustration of using (6) to sample a random solution $\boldsymbol{x} \sim \pi(\boldsymbol{\varphi}_{\boldsymbol{\theta}}(\boldsymbol{\alpha}))$.

As the Gaussian is smooth *w.r.t.* $\boldsymbol{\varphi}$, the network $\boldsymbol{\theta}$ can be trained by gradient ascent. A conceptual algorithm is inspired, which iteratively 1) sample $\lambda$ solutions $\{\boldsymbol{x}^j\}_{j=1}^{\lambda}$ from $\pi_{\boldsymbol{\theta}}(\cdot;\boldsymbol{\varphi})$ independently, and 2) estimate the gradient using these samples and back-propagate to $\boldsymbol{\theta}$ until convergence.

### 4.2 Implementation

#### 4.2.1 Rank-one covariance matrix

For any $i \in [p]$, a full covariance matrix $\boldsymbol{C}_{\boldsymbol{\theta}}(\boldsymbol{x}_{0:i-1})$ of $\Theta(d^2)$ degrees of freedom necessitates $\Theta(d^2)$ output neurons to generate it and $\Theta(d^2)$ time complexity to sample solutions as well as to estimate its gradient. To reduce the overall complexity to $\Theta(n)$, the above must be reduced to $\Theta(d)$.

As is a common trick in ES variants for high-dimensional tasks [43, 29, 30], we can replace a full covariance matrix by a *rank-one model*. More specifically, we define a step-size $\sigma_{\boldsymbol{\theta}}(\boldsymbol{x}_{0:i-1}) > 0$ and a component $\boldsymbol{v}_{\boldsymbol{\theta}}(\boldsymbol{x}_{0:i-1}) \in \mathbb{R}^d$, then $\boldsymbol{C}_{\boldsymbol{\theta}}(\boldsymbol{x}_{0:i-1})$ can be parameterized by:

$$\boldsymbol{C}_{\boldsymbol{\theta}}(\boldsymbol{x}_{0:i-1}) = \sigma_{\boldsymbol{\theta}}^2(\boldsymbol{x}_{0:i-1})\left(\boldsymbol{I} + \boldsymbol{v}_{\boldsymbol{\theta}}(\boldsymbol{x}_{0:i-1})\boldsymbol{v}_{\boldsymbol{\theta}}(\boldsymbol{x}_{0:i-1})^T\right). \qquad (7)$$

Of the above, $\sigma_{\boldsymbol{\theta}}$ controls the isotropic variance of the samples, and $\boldsymbol{v}_{\boldsymbol{\theta}}$ is expected to identify a promising search direction. Although the number of free parameters in $\boldsymbol{C}_{\boldsymbol{\theta}}(\boldsymbol{x}_{0:i-1})$ decreases, $\pi(\boldsymbol{\varphi})$ suffices to decouple complex dimensional dependencies when $p$ is large enough. By this means, the space complexity of $\boldsymbol{\theta}$'s output layer is reduced to $\Theta(d)$.

Following (7), a solution is practically generated by:

$$\boldsymbol{x}_i = \boldsymbol{m}_{\boldsymbol{\theta}}(\boldsymbol{x}_{0:i-1}) + \sigma_{\boldsymbol{\theta}}(\boldsymbol{x}_{0:i-1})(\boldsymbol{z} + r\boldsymbol{v}_{\boldsymbol{\theta}}(\boldsymbol{x}_{0:i-1})), \tag{8}$$

where $\boldsymbol{z} \sim \mathcal{N}(\boldsymbol{0}, \boldsymbol{I})$, $\boldsymbol{z} \in \mathbb{R}^d$, and $r \sim \mathcal{N}(0,1)$. After $p$ iterations, an $\boldsymbol{x} \sim \pi_{\boldsymbol{\theta}}(\cdot; \boldsymbol{\alpha})$ is recovered by stacking the $\boldsymbol{x}_1, ..., \boldsymbol{x}_p$. The time complexity of (8) is $\Theta(d)$.

### 4.2.2   Natural gradient

The gradient of $\boldsymbol{\theta}$ can be derived as

$$\nabla \bar{\mathcal{J}}_{\boldsymbol{\alpha}}(\boldsymbol{\theta}) = \sum_{i=1}^{p} \mathbb{E}_{\pi(\boldsymbol{x}; \boldsymbol{\varphi})} \left[ f(\boldsymbol{x}) \frac{\partial}{\partial \boldsymbol{\theta}} \left( \frac{\partial \ln \mathcal{N}(\boldsymbol{x}_i; \boldsymbol{\varphi}_{\boldsymbol{\theta}}(\boldsymbol{x}_{0:i-1}))}{\partial \boldsymbol{\varphi}_{\boldsymbol{\theta}}(\boldsymbol{x}_{0:i-1})} \right) \right]. \tag{9}$$

However, the partial derivative $\partial \ln \mathcal{N}(\boldsymbol{x}_i; \boldsymbol{\varphi}_{\boldsymbol{\theta}}(\boldsymbol{x}_{0:i-1})) / \partial \boldsymbol{\varphi}_{\boldsymbol{\theta}}(\boldsymbol{x}_{0:i-1})$ rarely yields stable convergence behaviors of the mean $\boldsymbol{m}_{\boldsymbol{\theta}}(\boldsymbol{x}_{0:i-1})$, because the shrinking of $\boldsymbol{C}_{\boldsymbol{\theta}}(\boldsymbol{x}_{0:i-1})$ in turns stretches the gradient of $\boldsymbol{m}_{\boldsymbol{\theta}}(\boldsymbol{x}_{0:i-1})$, preventing it from staying in stationary points. See [20, 26] as well as Section B.2 for more details.

To regulate the gradient, a remedy is to multiply each partial derivative in (9) by the inverse of the Fisher information matrix (a.k.a. the "Fisher") of $\boldsymbol{\varphi}_i = \boldsymbol{\varphi}_{\boldsymbol{\theta}}(\boldsymbol{x}_{0:i-1})$, which yields:

$$\tilde{\nabla} \bar{\mathcal{J}}_{\boldsymbol{\alpha}}(\boldsymbol{\theta}) := \sum_{i=1}^{p} \mathbb{E}_{\pi(\boldsymbol{x}; \boldsymbol{\varphi})} \left[ f(\boldsymbol{x}) \frac{\partial}{\partial \boldsymbol{\theta}} \left( \boldsymbol{\mathcal{F}}(\boldsymbol{\varphi}_i)^{-1} \frac{\partial \ln \mathcal{N}(\boldsymbol{x}_i; \boldsymbol{\varphi}_{\boldsymbol{\theta}}(\boldsymbol{x}_{0:i-1}))}{\partial \boldsymbol{\varphi}_{\boldsymbol{\theta}}(\boldsymbol{x}_{0:i-1})} \right) \right], \tag{10}$$

where $\boldsymbol{\mathcal{F}}(\boldsymbol{\varphi}_i) = \mathbb{E}_{\mathcal{N}(\boldsymbol{x}_i'; \boldsymbol{\varphi}_i)}[\nabla_{\boldsymbol{\varphi}_i} \ln \mathcal{N}(\boldsymbol{x}_i'; \boldsymbol{\varphi}_i) \nabla_{\boldsymbol{\varphi}_i} \ln \mathcal{N}(\boldsymbol{x}_i'; \boldsymbol{\varphi}_i)^T]$. The interpretation is that the Fisher penalizes drastic changes of the search distribution, counteracts the effect of the shrinking covariance, and thus guarantees stable convergence. More intrinsically, the following connects (10) to the natural gradient, which incorporates second-order information and is the steepest ascent direction on Riemannian manifolds [33].

**Lemma 1.** *Consider a distribution $\pi(\boldsymbol{\varphi})$ with fixed parameters $\boldsymbol{\varphi} = (\boldsymbol{\varphi}_1^T, ..., \boldsymbol{\varphi}_p^T)^T$ and a random sample $\boldsymbol{x} \sim \pi(\boldsymbol{\varphi})$. The partial derivative $\sum_{i=1}^{p}[\boldsymbol{\mathcal{F}}(\boldsymbol{\varphi}_i)^{-1} \partial \ln \mathcal{N}(\boldsymbol{x}_i; \boldsymbol{\varphi}_i)/\partial \boldsymbol{\varphi}_i]$ in (10) is an unbiased estimator for the natural gradient of the log-likelihood $\ln \pi(\boldsymbol{x}; \boldsymbol{\varphi})$ regarding $\boldsymbol{\varphi}$.*

Furthermore, empirical and theoretical evidence [20, 25, 26] have shown that unscaled objective values often lead to degenerated search distributions and premature convergence, due to the irregularity of $f$. ES algorithms alleviate this issue by replacing $f(\boldsymbol{x})$ with rank-dependent weight values. Specifically, consider a set of objective values $\{f(\boldsymbol{x}^j)\}_{j=1}^{\lambda}$ arranged in descending order, which means $f(\boldsymbol{x}^1) \geq ... \geq f(\boldsymbol{x}^\lambda)$, we replace each $f(\boldsymbol{x}^j)$ by $w_j$ where $w_1 \geq ... \geq w_\lambda$. In this paper, we define $w_j = \ln\left(\lfloor \frac{\lambda}{2} \rfloor + \frac{1}{2}\right) - \ln j$.

According to Akimoto *et al.* [40], the Fisher $\boldsymbol{\mathcal{F}}(\boldsymbol{\varphi}_i)$ can be absorbed into the gradient without explicitly computing its inverse. Eventually, the gradient estimator based on a set of solutions $\{\boldsymbol{x}^j\}_{j=1}^{\lambda}$ can be derived as:

$$\tilde{\nabla} \bar{\mathcal{J}}_{\boldsymbol{\alpha}}(\boldsymbol{\theta}) \approx \frac{1}{\lambda} \sum_{i=1}^{p} \sum_{j=1}^{\lambda} w_j \begin{bmatrix} \boldsymbol{J}(\boldsymbol{m}_{ij})^T(\boldsymbol{x}_i^j - \boldsymbol{m}_{ij}) + \\ 2\sigma_{ij}(\|\boldsymbol{y}_{ij}\|^2 + (\boldsymbol{v}_{ij}^T\boldsymbol{y})^2 - \sigma_{ij}^2(d + 2\|\boldsymbol{v}_{ij}\|^2 + \|\boldsymbol{v}_{ij}\|^4)) \frac{\partial \sigma_{ij}}{\partial \boldsymbol{\theta}} + \\ \sigma_{ij}^2 \boldsymbol{J}(\boldsymbol{v}_{ij})^T((\boldsymbol{v}_{ij}^T\boldsymbol{y}_{ij})\boldsymbol{y}_{ij} + \|\boldsymbol{y}_{ij}\|^2\boldsymbol{v} - 2\sigma_{ij}^2(1 + \|\boldsymbol{v}_{ij}\|^2)\boldsymbol{v}_{ij}) \end{bmatrix}, \tag{11}$$

where $\boldsymbol{m}_{ij} = \boldsymbol{m}_{\boldsymbol{\theta}}(\boldsymbol{x}_{0:i-1}^j)$, $\sigma_{ij} = \sigma_{\boldsymbol{\theta}}(\boldsymbol{x}_{0:i-1}^j)$, $\boldsymbol{v}_{ij} = \boldsymbol{v}_{\boldsymbol{\theta}}(\boldsymbol{x}_{0:i-1}^j)$, $\boldsymbol{y}_{ij} = \boldsymbol{x}_i^j - \boldsymbol{m}_{ij}$, $\boldsymbol{J}(\cdot)$ computes the $d \times |\boldsymbol{\theta}|$ dimensional Jacobian matrix of an input vector *w.r.t.* $\boldsymbol{\theta}$, and $\|\cdot\|$ computes the $\ell_2$-norm. We leave the tedious derivation steps in Section B.4. Each term irrelevant to $\boldsymbol{\theta}$ (i.e., those except $\boldsymbol{J}(\boldsymbol{m}_{ij})$, $\partial \sigma_{ij}/\partial \boldsymbol{\theta}$ and $\boldsymbol{J}(\boldsymbol{v}_{ij})$) yields $\Theta(d)$ time complexity.

### 4.3   The algorithm

We present Neural-ES in Alg. 1. During each epoch, i.e., lines 2 to 8, the algorithm optimizes $N$ randomly sampled subproblems $g(\cdot;\boldsymbol{\alpha}^k)$, $k \in [N]$. For each subproblem, lines 4 to 6 sample $\lambda$ random solutions from $\pi(\varphi_{\boldsymbol{\theta}}(\boldsymbol{\alpha}^k))$, and line 7 computes the gradient regarding this subproblem. Lines 4 to 7 correspond to an iteration of a typical ES algorithm. Line 8 estimates the gradient *w.r.t.* the PSL problem in (4), which reads $\tilde{\nabla}\mathcal{J}(\boldsymbol{\theta}) = \mathbb{E}_{\boldsymbol{\alpha}\sim\mathcal{U}(\Delta_M)}[\tilde{\nabla}\bar{\mathcal{J}}_{\boldsymbol{\alpha}^k}(\boldsymbol{\theta})] \approx \frac{1}{N}\sum_{k=1}^{N}\tilde{\nabla}\bar{\mathcal{J}}_{\boldsymbol{\alpha}^k}(\boldsymbol{\theta})$. Eventually, $\tilde{\nabla}\mathcal{J}(\boldsymbol{\theta})$ is back-propagated to $\boldsymbol{\theta}$, where $\eta$ denotes the learning rate.

**Algorithm 1:** Neural-ES for black-box PSL

**Input** : A black-box MOP $\boldsymbol{F}(\boldsymbol{x})$
**Output** : A set model $\varphi_{\boldsymbol{\theta}}$

```
1 while The termination is not triggered do
2     for k = 1 to N do
3         Sample a preference α^k ~ U(Δ_M), set x_0 = α^k;
4         for j = 1 to λ do
5             for i = 1 to p do
6                 Construct x_i^j based on (8);
7         Compute ∇̃J̄_{α^k}(θ) based on (11), where f(·) := −g(·; α^k);
8     θ ← θ + (η/N) Σ_{k=1}^{N} ∇̃J̄_{α^k}(θ);
```

# 5 Experiments

## 5.1 Configurations

**Black-Box Pareto Set Learning (BBO-PSL) suite** To evaluate the performance of PSL algorithms, a test problem should meet a few fundamental requirements, including 1) closed-form PF and PS, 2) a complex shape of the PS manifold, 3) scalability to arbitrarily high dimensions, and 4) non-separability with a wide spectrum of the Hessian. However, most existing problem suites [44–47] either implement only a few of these requirements or include other interfering features that fall outside the scope of this paper. To this end, we present the novel BBO-PSL suite, which comprises eight instances (F1 to F8) and satisfies all the aforementioned requirements. Among them, F1 to F4 are bi-objective instances, and F5 to F8 are tri-objective.

**Compared algorithms** We compare Neural-ES [2] to the state-of-the-art and canonical EPSL [15], and configure Neural-ES by $d = \lfloor n/16 \rceil$, $N = \lfloor 5n/16 \rceil$, and $\lambda = 2 + \lfloor 1.5 \ln n \rceil$. Additionally, as MOEAs are widely recognized as one of the most effective methodologies for obtaining a set of non-dominated solutions, we consider two state-of-the-art MOEAs specifically designed for high-dimensional MOPs, namely LMO-CSO [35] and IM-MOEA/D [36]. Among the many MOEAs, LMO-CSO is favored for its superior handling of dimensional dependencies in complex and high-dimensional landscapes, while IM-MOEA/D makes a notable effort to elevate the canonical MOEA/D [8] to high dimensions by incorporating the popular inverse modeling technique [48]. Compared to most others, they are more lightweight with lower computational overhead, indicating better scalability to larger population sizes. Moreover, we synthesize an algorithm called EPSL-R1, by setting $p = 1$ in Neural-ES, which represents the simplest way to integrate PSL with an ES.

**Metrics** The Inverted Generational Distance (IGD) [49] is used as the performance metric, with a smaller value indicating better performance. Each algorithm is repeated 21 times on each BBO-PSL instance using different random seeds. For the three PSL algorithms, we evaluate their *testing performance*. To do so, we input a large number of preferences into a trained set model, which yields the same number of distributions; from each distribution, we randomly sample a solution, the collection of which form the *testing solution set*; finally, we measure the IGD value of the set and refer it as the *testing IGD value*. Meanwhile, we track the convergence of the PSL algorithms during training, in terms of the IGD values acquired by the best solutions so far. The MOEAs directly optimize the

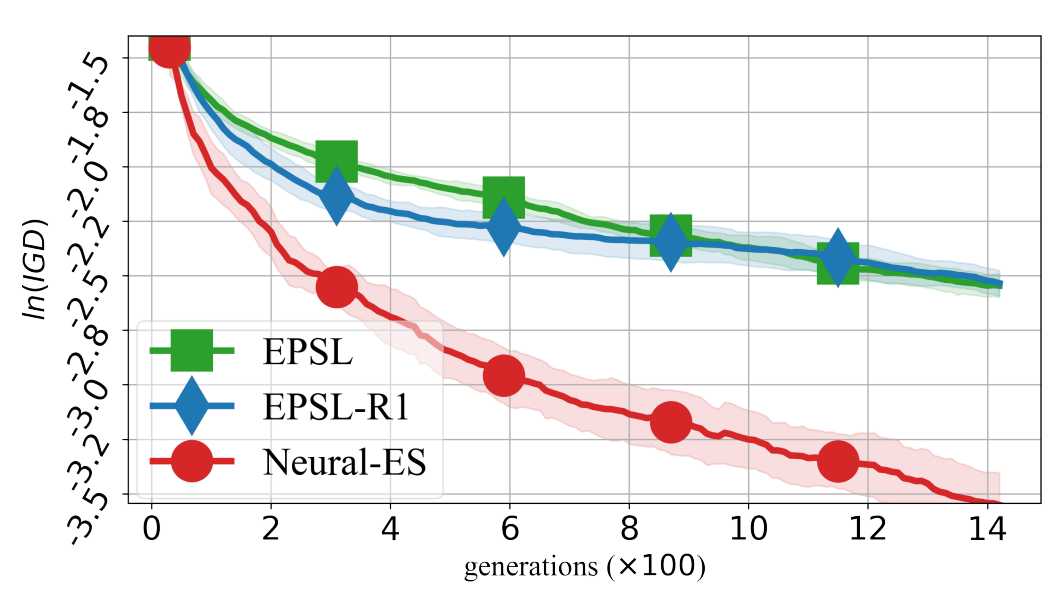

Figure 3: Convergence of the PSL algorithms on F1-128D.

[2] Code of Neural-ES and the BBO-PSL suite is available in `https://github.com/chandler09/Neural-Evolution-Strategy`.

same number of solutions, and the IGD value of the final population is measured. To examine the scalability of the algorithms with increasing dimensions, we vary $n$ from 32 to 1024.

## 5.2 Results and discussions

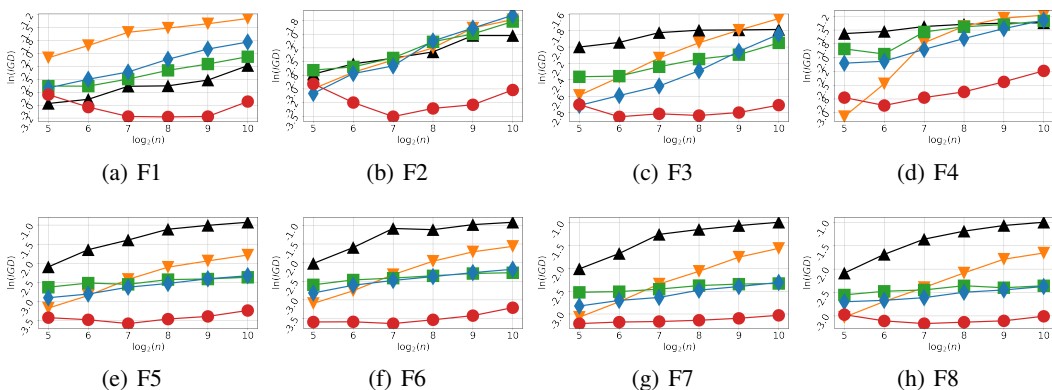

(a) F1   (b) F2   (c) F3   (d) F4

(e) F5   (f) F6   (g) F7   (h) F8

Figure 4: Average testing IGD values *w.r.t.* problem dimensions, where the red curve with circles refers to Neural-ES (ours), the blue curve with diamonds refers to EPSL-R1, the green curve with squares refers to EPSL, the orange curve with upside-down triangles refers to LMO-CSO, and finally the black curve with triangles refers to IM-MOEA/D. Their detailed values, along with the Mann-Whitney U test, can be found in Table 3.

According to Fig. 4, Neural-ES outperforms all the other algorithms in all eight instances by a large margin. Although the MOEAs are specialized for high-dimensional MOPs, they struggle to scale efficiently when users require numerous trade-offs among objectives. This drawback arises from their substantial demand on evaluation budgets and limited generalization ability. In contrast, the PSL paradigm performs well during surges in user demand, especially when demands shift or new ones emerge after optimization begins. The PSL algorithms, and Neural-ES in particular, are more robust against increasing problem dimensions. All these are attributed to the efficiency of evolution strategies and the generalization ability of the set model.

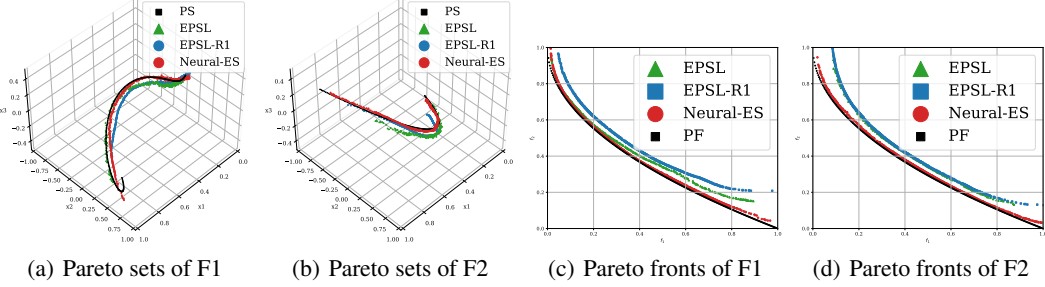

(a) Pareto sets of F1   (b) Pareto sets of F2   (c) Pareto fronts of F1   (d) Pareto fronts of F2

Figure 5: Testing solution sets of the algorithms, on F1-512D and F2-512D.

Compared to EPSL and EPSL-R1, Neural-ES reduces the IGD values by effectively decoupling the dimensional dependencies. Although EPSL-R1 also handles the dependencies, it performs slightly better than EPSL in just a few cases, yet remains similar overall. And its advantage over EPSL dissipates as $n$ increases. This verifies that simply plugging an ES algorithm into adaptation-free scenarios, such as black-box PSL, does not yield a competitive algorithm. In contrast, the novel treatment of dimensional dependencies positions Neural-ES as a highly efficient alternative in adaptation-free scenarios.

In the remainder, we limit our focus to the PSL paradigm and utilize the limited space for in-depth discussions. In addition to the testing performance, we plot in Fig. 3 the training-time convergence of the three PSL algorithms. Neural-ES establishes a performance lead over EPSL and EPSL-R1

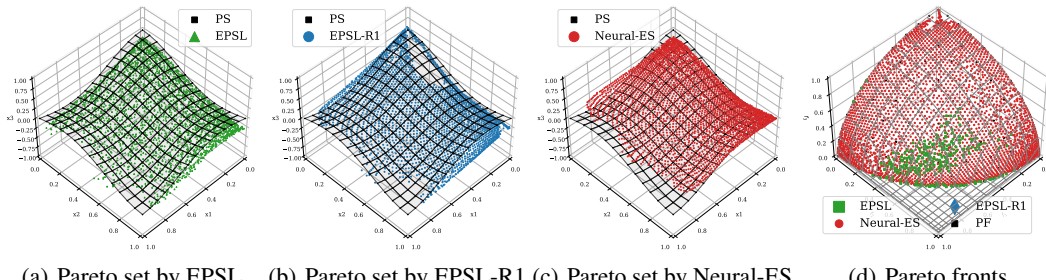

(a) Pareto set by EPSL   (b) Pareto set by EPSL-R1 (c) Pareto set by Neural-ES   (d) Pareto fronts

Figure 6: Testing solution sets of the algorithms, on F5-128D. For the ease of visualization, we only demonstrate the non-dominated solutions in 6(d). Therefore, the results of EPSL-R1 are unreported because they were dominated.

during the early stages, and subsequently extends the performance gap throughout the search process. The results suggest that Neural-ES efficiently learns the complex dimensional dependencies from the outset and exploits this advantage to significantly reduce the optimization difficulty.

The ability to approximate Pareto manifolds during testing is a unique feature of PSL as well as its primary priority. We plot in Figs. 5 and 6 some example approximation results. For the bi-objective F1 and F2, in Figs. 5(a) and 5(b), the approximate PSs by Neural-ES align more closely with the ground-truth and spread much wider than EPSL and EPSL-R1. It turns out that the approximate PFs by Neural-ES dominate those by the other two almost everywhere, as seen in Figs. 5(c) and 5(d).

Figs. 6(a) to 6(c) depict the approximate PSs in the first three dimensions of the search space (i.e., the $x_1 - x_2 - x_3$ subspace of the $x$-space). In these figures, the gray surface represents the ground-truth PS, while the colored points indicate the sampled solutions obtained by different algorithms during testing. In Fig. 6(a), solutions sampled by EPSL appear scattered and poorly aligned with preference vectors, because 1) the finite difference approach does not minimize the uncertainty of a distribution, and 2) EPSL does not learn the dimensional dependencies. Fig. 6(d) depicts the ground-truth PF (i.e., surface with gray contours) and the algorithm approximations in the objective space (i.e., the $F$-space), which provides a more rigorous performance assessment because it directly evaluates solution quality in the objective space and reveals dominance relationships that are not apparent in the $x$-space. While Neural-ES and EPSL-R1 appear similar in decision space (Figs. 6(a) to 6(c)), Neural-ES clearly outperforms in objective space, with most EPSL-R1 solutions being dominated by Neural-ES solutions.

## 5.3 Real-world applications

**Trajectory planning** We consider a tri-objective trajectory planning task modified from the popular rover problem [50, 34]. The goal is to determine a trajectory that minimizes 1) the cost during navigation, e.g., collisions with obstacles, 2) the distance from the endpoint to an intended target location, and 3) the total trajectory length, e.g., for saving energy. The objectives conflict with each other because, for instance, minimizing the distance may lead to increased frequency of collisions. We plan the trajectory by designing a total of 50 (x,y) coordinates, which means $n = 100$. Their approximate PFs are depicted in Fig. 7. Measured at a reference point of (5,2,1.5), Neural-ES, EPSL-R1, and EPSL achieve an average hypervolume of $8.314\pm0.215$, $5.144\pm0.622$, and $4.186\pm0.709$, respectively, with higher values indicating better performance. Additionally, for each algorithm, we collect the optimal solutions for 300 evenly distributed subproblems. 299 out of 300 solutions from EPSL were dominated by those from Neural-ES, while 194 from EPSL-R1 were also dom-

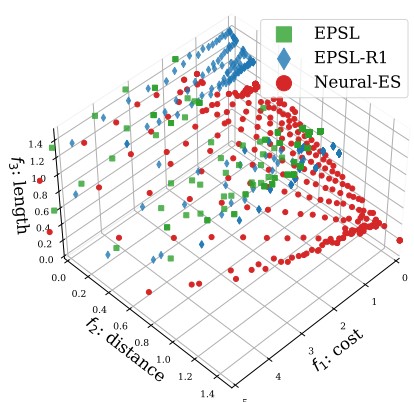

Figure 7: Approximate PFs of the trajectory planning problem.

inated by those from Neural-ES. The results demonstrate
the superiority of Neural-ES in balancing the cost, distance, and total length of the trajectories.

**Multi-objective Unmanned Aerial Vehicle (MO-UAV) Navigation** We consider a set of real-world multi-objective MO-UAV navigation problems, which are featured in the Meta-BBO-v2 library [51] [3]. This problem consists of three competing objectives: 1) the distance from the UAV's final location to the destination, penalized

Table 1: Average HV values and standard deviations, on the MO-UAV problems, with larger values the better.

| Case | $n$ | EPSL | EPSL-R1 | Neural-ES |
|------|-----|------|---------|-----------|
| Easy | 30 | $2.47_{\pm 0.36}$ | $1.58_{\pm 0.15}$ | $\mathbf{3.40}_{\pm 0.28}$ |
| Moderate | 60 | $2.40_{\pm 0.37}$ | $1.52_{\pm 0.27}$ | $\mathbf{3.44}_{\pm 0.45}$ |
| Hard | 120 | $0.72_{\pm 0.20}$ | $0.84_{\pm 0.14}$ | $\mathbf{2.44}_{\pm 1.14}$ |

by the trajectory's non-smoothness, 2) the risk of collision with threats, such as ground-to-air missiles, and 3) an altitude penalty to maintain safe and cost-efficient flight. By varying the control parameters of the terrain, we have developed three instances: easy, moderate, and hard, which require a UAV trajectory of 10, 20, and 40 waypoints, respectively. Because each point comprises $x$, $y$, and $z$ coordinates, the overall problem dimensions are 30, 60, and 120, respectively. Complex dimensional dependencies arise from the spatial and temporal correlations between waypoints, making navigation highly challenging. After 21 independent runs, we report the brief results in Table 1. Neural-ES demonstrates its superiority in practice with the highest HV values across all difficulty levels.

## 6 Conclusion, limitation, and future work

**Conclusions** In this work, we have made an initial effort to solve a novel yet important problem: black-box Pareto set learning (PSL). The goal is to learn the entire Pareto set of a black-box multi-objective function. We have developed a novel formulation, which facilitates the use of powerful black-box optimization algorithms, particularly evolution strategies. To address the challenges posed by black-box PSL, we have developed a novel mechanism to decouple the complex dimensional dependencies of the objectives. The proposed Neural-ES excels at learning challenging Pareto sets, successfully scaling the PSL paradigm to over 1000 dimensions with competitive performance.

**Limitation and future work** While this work focuses on unconstrained MOPs, constraints often significantly increase optimization difficulties, making constrained MOPs a critical area to address in PSL. Developing novel and advanced methods to extend PSL to constrained MOPs is a promising yet challenging direction for future research. Besides, Ye *et al.* [7] recently suggest that when reparameterizing an aggregation function $g(\boldsymbol{x}; \boldsymbol{\alpha})$ as $\mathbb{E}_{\pi(\boldsymbol{x}; \boldsymbol{\varphi})}[g(\boldsymbol{x}; \boldsymbol{\alpha})]$, the optimization may only guarantee an approximate Pareto stationary point rather than an exact one, in regardless of the specific distribution $\pi$. It is very interesting to further investigate what led to such a mismatch. Nonetheless, we believe that the formulation in (4) remains one of the best options available.

## Acknowledgments

The work described in this paper was supported by the Research Grants Council of the Hong Kong Special Administrative Region, China [GRF Project No. CityU-11215723] and the Natural Science Foundation of China (Project No: 62276223). Zhenhua Li is supported in part by the Fundamental Research Funds for the Central Universities under No.NJ2024031.

---

[3] https://github.com/MetaEvo/MetaBox

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

# A  Additional related works

## A.1  Evolution strategies

ES [20, 25, 21] is a prominent class of second-order black-box optimization methods. Compared to zeroth-order methods that leverage probability distribution models in their simplest forms and usually amount to finite-difference approaches [52], ES is considered second-order because they are closely related to the quasi-Newton method [53]: they typically learn a covariance matrix that converges to the inverse of the objective's Hessian.

Wierstra *et al.* [20] proposed the Natural Evolution Strategy (NES), which amounts to a stochastic natural gradient ascent. Since its publication, several NES variants have emerged [54, 28, 43], addressing the issue with the Fisher information matrix using different parameterization strategies. Both Akimoto *et al.* [40] and Glasmachers *et al.* [28] derived the rank-$\mu$ update of CMA-ES from natural gradient, bridging gradient search and state-of-the-art ES. The primary difference between CMA-ES and NES, which makes the former superior, is that CMA-ES leverages two evolution paths that act as the momentum term in gradient search [55], one for updating the covariance matrix, while the other for tuning the isotropic mutation strength.

The covariance matrix of the Gaussian distribution is one of the most contributing parameters of ES. Through extensive experiments, Hansen *et al.* [24] showed that with the help of an adaptive covariance matrix, CMA-ES is not affected by coordinate system rotations, and is more resistant to the increase of the condition number. Beyer [26] proved that the covariance matrix gets asymptotically proportional to the inverse of the objective's Hessian, which is preferable as it counteracts the degeneration tendencies of the search distribution into a subspace.

The $\Theta(n^2)$ space and time complexities regarding the covariance matrix hinder the application of ES to large-scale problems. Several methods have been proposed to remedy the issue. These methods agree on simplifying the covariance matrix, while differing in the details of designing and updating the matrix. Sun *et al.* [43] proposed R1-NES, where a rank-one model is adopted as the covariance matrix, and the update is completely derived from natural gradient ascent. Akimoto *et al.* [29] proposed VD-CMA, where the covariance matrix is parameterized by the multiplication of a rank-one model, and a diagonal matrix that captures the scaling of each dimension. They derived the update from natural gradient and, in the meantime, incorporated the cumulation techniques of evolution paths from CMA-ES. The experiments demonstrated that VD-CMA significantly outperformed R1-NES. The multiplicative model in VD-CMA is further pursued in dd-CMA [56], a recent method that can well adapt to both separable and non-separable scenarios. A straightforward extension to the rank-one model is to apply more than one component to capture multiple promising search directions simultaneously, which yields much more efficient algorithms [57, 30, 31].

Although some recently proposed ESs are versatile across various tasks [23, 22], it may be more appropriate to classify them as zeroth-order methods, considering that they mainly focus on reducing the variance of the finite difference without utilizing second-order information. On the other hand, while some other methods do not explicitly identify as ESs [58, 59], they are to some extent inspired by ESs and share many common principles. Notably, Lyu and Tsang [58] proposed a black-box optimization algorithm called Implicit Natural Gradient Optimization (INGO). To avoid explicitly inverting the Fisher, they formulated a surrogate objective where the KL-divergence is attached as a penalty term and can be explicitly derived. The authors showed that the gradient with respect to the expectation of the sufficient statistics amounts to the natural gradient of the natural parameter of an exponential-family distribution. Since the Hessian of an objective primarily characterizes the dimensional dependencies, we focus exclusively on second-order methods.

## A.2  Black-box multi-objective optimization

Multi-objective optimization (MOO) has consistently garnered significant attention [34, 7, 34]. In black-box MOO, multi-objective evolutionary algorithms (MOEAs) remain among the most dominant methodologies [6, 8, 60]. This is because MOEAs excel in maintaining a large population of solutions, and can thus approach the underlying Pareto Front (PF) from all directions. Recently, dedicated efforts have been made to extend MOEAs to high-dimensional, or in other words, large-scale MOPs. For instance, IM-MOEA/D [36] integrates the canonical MOEA/D [5] with an inverse model [48] which directly samples in the objective space using Gaussian process models, and then maps into the decision space. Focusing on the complex landscapes in high-dimensional MOPs, LMO-CSO [35] proposes addressing the dimensional dependencies using an efficient particle swarm optimization (PSO) algorithm. Among the three mainstream MOEAs, those based on Pareto dominance and performance indicators fall short of aligning with decision-makers' preferences and scale poorly with the number of solutions [60], due to the high computational overhead associated with non-dominated sorting and hypervolume contribution. Meanwhile, decomposition-based MOEAs [5] struggle with only a limited number of pre-established preferences, generalize poorly to the entire Pareto manifold, and encounter difficulties in sharing information across subproblems.

# B  More details about Neural-ES

This section is arranged as follows:

- Section B.1 shows that random samples generated by (8) are in-distribution of $\pi(\cdot; \boldsymbol{\varphi})$, where $\boldsymbol{\varphi} = \boldsymbol{\varphi}_{\boldsymbol{\theta}}(\boldsymbol{\alpha})$;
- Section B.2 derives the vanilla gradient $\nabla \bar{\mathcal{J}}_{\boldsymbol{\alpha}}(\boldsymbol{\theta})$ in (9), and analyzes its major drawback, that is, it potentially leads to divergent behaviors of the search distribution.
- Section B.3 justifies the use of the Fisher from the perspective of natural gradient, and derives (10).
- Section B.4 further transforms the gradient (10) to the practical update rule (11) used in the algorithm.
- Sections B.5 and B.6 provide more details on the weight values and the set model.
- Section B.7 derives the space and time complexity of Neural-ES, demonstrating its time efficiency.

## B.1  Proof of Equation (8)

**Proposition 1.** *A solution generated by (8) follows the distribution $\pi(\cdot; \boldsymbol{\varphi})$, where $\boldsymbol{\varphi} = \boldsymbol{\varphi}_{\boldsymbol{\theta}}(\boldsymbol{\alpha})$.*

*Proof.* Consider the solution $\boldsymbol{x} = (\boldsymbol{x}_1, ..., \boldsymbol{x}_p)$ where $\boldsymbol{x}_i = \boldsymbol{m}_{\boldsymbol{\theta}}(\boldsymbol{x}_{0:i-1}) + \sigma_{\boldsymbol{\theta}}(\boldsymbol{x}_{0:i-1})(\boldsymbol{z} + r\boldsymbol{v}_{\boldsymbol{\theta}}(\boldsymbol{x}_{0:i-1}))$ for all $i \in [p]$, $\boldsymbol{z} \sim \mathcal{N}(\boldsymbol{0}, \boldsymbol{I})$, and $r \sim \mathcal{N}(0, 1)$. To prove $\boldsymbol{x} \sim \pi(\boldsymbol{\varphi})$ is equivalent to proving that $(\boldsymbol{x}_1, ..., \boldsymbol{x}_p)$ follows the joint distribution $\prod_{i=1}^{p} \mathcal{N}_{\boldsymbol{\theta}}(\cdot | \boldsymbol{x}_0, \boldsymbol{x}_1, ..., \boldsymbol{x}_{i-1})$ where $\boldsymbol{x}_0 = \boldsymbol{\alpha}$.

Note that for an arbitrary $i \in [p]$, the expectation of $\boldsymbol{x}_i$ is given by:

$$\mathbb{E}[\boldsymbol{x}_i] = \boldsymbol{m}_{\boldsymbol{\theta}}(\boldsymbol{x}_{0:i-1}) + \sigma_{\boldsymbol{\theta}}(\boldsymbol{x}_{0:i-1}) \left( \underbrace{\mathbb{E}[\boldsymbol{z}]}_{=\boldsymbol{0}} + \underbrace{\mathbb{E}[r]}_{=0} \boldsymbol{v}_{\boldsymbol{\theta}}(\boldsymbol{x}_{0:i-1}) \right) = \boldsymbol{m}_{\boldsymbol{\theta}}(\boldsymbol{x}_{0:i-1}),$$

and the covariance matrix is given by:

$$\mathbb{E}\left[(\boldsymbol{x}_i - \boldsymbol{m}_{\boldsymbol{\theta}}(\boldsymbol{x}_{0:i-1}))(\boldsymbol{x}_i - \boldsymbol{m}_{\boldsymbol{\theta}}(\boldsymbol{x}_{0:i-1}))^T\right]$$

$$= \sigma_{\boldsymbol{\theta}}(\boldsymbol{x}_{0:i-1})^2 \left( \underbrace{\mathbb{E}[\boldsymbol{z}\boldsymbol{z}^T]}_{=\boldsymbol{I}} + \boldsymbol{v}_{\boldsymbol{\theta}}(\boldsymbol{x}_{0:i-1})\mathbb{E}[r\boldsymbol{z}]^T + \mathbb{E}[r\boldsymbol{z}]\boldsymbol{v}_{\boldsymbol{\theta}}(\boldsymbol{x}_{0:i-1})^T + \underbrace{\mathbb{E}[r^2]}_{=1} \boldsymbol{v}_{\boldsymbol{\theta}}(\boldsymbol{x}_{0:i-1})\boldsymbol{v}_{\boldsymbol{\theta}}(\boldsymbol{x}_{0:i-1})^T \right)$$

$$= \sigma_{\boldsymbol{\theta}}(\boldsymbol{x}_{0:i-1})^2 \left( \boldsymbol{I} + \boldsymbol{v}_{\boldsymbol{\theta}}(\boldsymbol{x}_{0:i-1})\boldsymbol{v}_{\boldsymbol{\theta}}(\boldsymbol{x}_{0:i-1})^T \right) = \boldsymbol{C}_{\boldsymbol{\theta}}(\boldsymbol{x}_{0:i-1}),$$

which is because $\mathbb{E}[r\boldsymbol{z}] = \mathbb{E}[r]\mathbb{E}[\boldsymbol{z}] = \boldsymbol{0}$, as $r$ and $\boldsymbol{z}$ are independent from each other. Therefore, for all $i \in [p]$, $\boldsymbol{x}_i \sim \mathcal{N}(\boldsymbol{m}_{\boldsymbol{\theta}}(\boldsymbol{x}_{0:i-1}), \boldsymbol{C}_{\boldsymbol{\theta}}(\boldsymbol{x}_{0:i-1}))$, and $\mathcal{N}(\boldsymbol{m}_{\boldsymbol{\theta}}(\boldsymbol{x}_{0:i-1}), \boldsymbol{C}_{\boldsymbol{\theta}}(\boldsymbol{x}_{0:i-1})) = \mathcal{N}_{\boldsymbol{\theta}}(\cdot | \boldsymbol{x}_0, \boldsymbol{x}_1, ..., \boldsymbol{x}_{i-1})$.

By the definition of joint distribution, we have:

$$(\boldsymbol{x}_1, \boldsymbol{x}_2) \sim \mathcal{N}_{\boldsymbol{\theta}}(\cdot | \boldsymbol{x}_0)\mathcal{N}_{\boldsymbol{\theta}}(\cdot | \boldsymbol{x}_0, \boldsymbol{x}_1),$$

$$(\boldsymbol{x}_1, \boldsymbol{x}_2, \boldsymbol{x}_3) \sim \mathcal{N}_{\boldsymbol{\theta}}(\cdot | \boldsymbol{x}_0)\mathcal{N}_{\boldsymbol{\theta}}(\cdot | \boldsymbol{x}_0, \boldsymbol{x}_1)\mathcal{N}_{\boldsymbol{\theta}}(\cdot | \boldsymbol{x}_0, \boldsymbol{x}_1, \boldsymbol{x}_2),$$

$$\vdots$$

$$(\boldsymbol{x}_1, \boldsymbol{x}_2, ..., \boldsymbol{x}_p) \sim \prod_{i=1}^{p} \mathcal{N}(\cdot | \boldsymbol{x}_0, ..., \boldsymbol{x}_{i-1}).$$

$\square$

By this means, we show that *the reparameterization trick $\boldsymbol{x}_i = \boldsymbol{m}_{\boldsymbol{\theta}}(\boldsymbol{x}_{0:i-1}) + \sigma_{\boldsymbol{\theta}}(\boldsymbol{x}_{0:i-1})(z + r\boldsymbol{v}_{\boldsymbol{\theta}}(\boldsymbol{x}_{0:i-1}))$ in (8) correctly generate in-distribution samples $\boldsymbol{x} \sim \pi(\boldsymbol{\varphi})$.*

## B.2  Issue with (9): a divergent gaussian mean

**Deriving (9).**  Recall that (9) presents the vanilla gradient of $\bar{\mathcal{J}}_{\boldsymbol{\alpha}}(\boldsymbol{\theta}) = \mathbb{E}_{\pi(\boldsymbol{x};\boldsymbol{\varphi})}[f(\boldsymbol{x})]$ w.r.t. $\boldsymbol{\theta}$, which is expressed as:

$$\nabla \bar{\mathcal{J}}_{\boldsymbol{\alpha}}(\boldsymbol{\theta}) = \sum_{i=1}^{p} \mathbb{E}_{\pi(\boldsymbol{x};\boldsymbol{\varphi})} \left[ f(\boldsymbol{x}) \frac{\partial}{\partial \boldsymbol{\theta}} \left( \frac{\partial \ln \mathcal{N}(\boldsymbol{x}_i; \boldsymbol{\varphi}_i)}{\partial \boldsymbol{\varphi}_i} \right) \right],$$

where $\varphi_i = \varphi_\theta(x_{0:i-1})$. We are going to derive the term $\frac{\partial \ln \mathcal{N}(x_i; \varphi_i)}{\partial \varphi_i}$. Notice that the partial derivative of $\pi(x; \varphi) = \prod_{i=1}^{p} \mathcal{N}(x_i; \varphi_i)$ w.r.t. $\varphi$ can be derived as:

$$\frac{\partial \ln \pi(x; \varphi)}{\partial \varphi} = \frac{1}{\pi(x; \varphi)} \frac{\partial \pi(x; \varphi)}{\partial \varphi} = \sum_{i=1}^{p} \frac{\partial \ln \mathcal{N}(x_i; \varphi_i)}{\partial \varphi_i}, \Rightarrow \frac{\partial \pi(x; \varphi)}{\partial \varphi} = \pi(x; \varphi) \sum_{i=1}^{p} \frac{\partial \ln \mathcal{N}(x_i; \varphi_i)}{\partial \varphi_i}. \tag{12}$$

Using the above, we can derive the partial derivatives $\partial \bar{\mathcal{J}} / \partial \varphi$:

$$\frac{\partial \bar{\mathcal{J}}}{\partial \varphi_i} = \int_x f(x) \frac{\partial \pi(x; \varphi)}{\partial \varphi} dx = \int_x f(x) \pi(x; \varphi) \sum_{i=1}^{p} \frac{\partial \ln \mathcal{N}(x_i; \varphi_i)}{\partial \varphi_i} dx = \sum_{i=1}^{p} \mathbb{E}_{\pi(x; \varphi)} \left[ f(x) \frac{\partial \ln \mathcal{N}(x_i; \varphi_i)}{\partial \varphi_i} \right]. \tag{13}$$

Eventually, the gradient w.r.t. $\theta$ can be derived by directly applying the chain rule, which recovers (9).

**The issue.** Regarding a random sample $x_i \in \mathbb{R}^d$, we refer to $\partial \ln \mathcal{N}(x_i; \varphi_i) / \partial \varphi_i$ in (13) as the vanilla gradient of the log-likelihood of $\mathcal{N}(x_i; \varphi_i)$. The biggest issue is perhaps that it potentially hinders the convergence of $m_i$, which explains why 1) vanilla gradient is not popular and 2) some so-called ES variants proposed recently [23, 22] either update the distribution mean only or update the covariance matrix using gradient-free strategies. For an arbitrary $\mathcal{N}(x_i; \varphi_i)$ where $\varphi_i^T = (m_i^T, \text{vec}(C_i)^T)$, and vec $\cdot$ is the vectorization operator that stacks all the columns of an input matrix [61]. With a slight abuse of symbols, we consider in this and the next subsections that the mean $m_i$ and covariance matrix $C_i$ are constant parameters. The vanilla gradient regarding a random sample $x_i$ can be derived as [40]:

$$\frac{\partial \ln \mathcal{N}(x_i; \varphi_i)}{\partial \varphi_i} = \begin{pmatrix} \frac{\partial \ln \mathcal{N}(x_i; \varphi_i)}{\partial m_i} \\ \frac{\partial \ln \mathcal{N}(x_i; \varphi_i)}{\partial \text{vec}(C_i)} \end{pmatrix} = \begin{pmatrix} C_i^{-1}(x_i - m_i) \\ \frac{1}{2} \text{vec}\left( C_i^{-1}(x_i - \varphi_i)(x_i - \varphi_i)^T C_i^{-1} - C_i^{-1} \right) \end{pmatrix}. \tag{14}$$

By eigenvalue decomposition, $C_i = Q \Lambda \Lambda Q^T$, where $\Lambda$ is a diagonal matrix whose main diagonal consists of the square roots of $C_i$'s eigenvalues, and $Q$ is an orthonormal matrix satisfying $Q^T = Q^{-1}$. $x_i$ can thus be reparameterized by $x_i = m_i + Q \Lambda z_i$ where $z_i \sim \mathcal{N}(0, I)$. Consequently, we can rewrite

$$\frac{\partial \ln \mathcal{N}(x_i; \varphi_i)}{\partial m_i} = \overbrace{Q^{-T} \Lambda^{-1} \Lambda^{-1} Q^{-1}}^{=C_i^{-1}} (\overbrace{m_i + Q \Lambda z_i}^{=x_i} - m_i) = \overbrace{Q^{-T}}^{=Q} \Lambda^{-1} \overbrace{\Lambda^{-1} Q^{-1} Q \Lambda}^{=I} z_i = Q \Lambda^{-1} z_i, \tag{15}$$

whose variance is:

$$\text{Var}\left( \frac{\partial \ln \mathcal{N}(x_i; \varphi_i)}{\partial m_i} \right) = \mathbb{E}\left[ (z_i - \overbrace{\mathbb{E}[z_i]}^{=0})^T \Lambda^{-1} \overbrace{Q^T Q}^{=I} \Lambda^{-1} (z_i - \mathbb{E}[z_i]) \right] = \sum_{j=1}^{d} \frac{1}{\nu_j} \overbrace{\mathbb{E}[z_j^2]}^{=1} = \text{Tr}(C_i^{-1}), \tag{16}$$

where $z_i = (z_1, ..., z_d)^T$, $\nu_i$ is the $i$-th largest eigenvalue of $C_i$, $\mathbb{E}[z^j] = 1$ for all $j \in [d]$ is due to the mean of the chi-squared distribution, and $\text{Tr}(\cdot)$ denotes the trace of a matrix. Intuitively, during the search process, *the entropy of the search distribution, i.e., the (log-)determinant of $C_i$, usually decreases, which is likely to increase* $\text{Tr}(C_i^{-1})$, *and therefore prevents $m_i$ from staying in a stationary point.* The phenomenon has been observed in [20] and rigorously discussed in [26].

## B.3 Proof of Lemma 1 & interpretation of (10)

Define $\pi(x; \varphi) = \prod_{i=1}^{p} \mathcal{N}(x_i; \varphi_i)$ and $\varphi_i = (m_i^T, \text{vec}\, C_i^T)^T$. Note that we treat $m_i$ and $C_i$ as constant parameters instead of mappings from $x_{0:i-1}$. We introduce the definition of natural gradient and its geometric interpretation:

**Definition 4** (Natural gradient [32]). Consider a distribution $\pi(\cdot; \varphi)$ that is differentiable w.r.t. its parameters $\varphi$, with the p.d.f. $\pi(x; \varphi)$. The natural gradient of any differentiable function $h(\varphi)$ w.r.t. $\varphi$ is given by

$$\tilde{\nabla} h(\varphi) = \mathcal{F}(\varphi)^{-1} \nabla h(\varphi), \tag{17}$$

where and $\mathcal{F}(\varphi) = \mathbb{E}_{\pi(x; \varphi)}[\nabla \ln \pi(x; \varphi) \nabla \ln \pi(x; \varphi)^T]$ is referred to as the Fisher information matrix w.r.t. $\varphi$.

**Proposition 2** (Geometric interpretation of natural gradient [25]). *The natural gradient $\tilde{\nabla} h(\varphi)$ points in the direction $\delta$ that yields the greatest improvement of h, for a given distance between $\pi(\cdot; \varphi)$ and $\pi(\varphi + \delta)$ in Kullback-Leibler divergence. More precisely, let $\varphi \in \Phi$ be a point where $\|\tilde{\nabla} h(\varphi)\| > 0$, then*

$$\delta^* = \frac{\tilde{\nabla} h(\varphi)}{\|\tilde{\nabla} h(\varphi)\|} = \lim_{\epsilon \to 0^+} \frac{1}{\epsilon} \underset{\delta}{\text{argmax}}\; h(\varphi + \delta), \text{ s. t. } \mathcal{KL}\left( \mathcal{N}(\pi(\varphi)) \,\|\, \mathcal{N}(\pi(\varphi + \delta)) \right) \leq \frac{\epsilon^2}{2}. \tag{18}$$

Of the above, **definition** 4 provides the form of a natural gradient, and **proposition** 2 suggests that a natural gradient identities the steepest ascent direction on Riemannian manifolds, and in the meanwhile, guarantees steady changes of $\pi$.

We prove Lemma 1 as follows:

*Proof.* First, the gradient of $\ln \pi(\boldsymbol{x}; \boldsymbol{\varphi})$ is:

$$\nabla \ln \pi(\boldsymbol{x}; \boldsymbol{\varphi}) = \frac{\partial \ln \pi(\boldsymbol{x}; \boldsymbol{\varphi})}{\partial \boldsymbol{\varphi}} = \left( \frac{\partial \ln \mathcal{N}(\boldsymbol{x}_1; \boldsymbol{\varphi}_1)}{\partial \boldsymbol{\varphi}_1}^T, ..., \frac{\partial \ln \mathcal{N}(\boldsymbol{x}_p; \boldsymbol{\varphi}_p)}{\partial \boldsymbol{\varphi}_p}^T \right)^T.$$

According to the definition of the Fisher,

$$\boldsymbol{\mathcal{F}}(\boldsymbol{\varphi}) = \mathbb{E}_{\pi(\boldsymbol{x}; \boldsymbol{\varphi})} \begin{bmatrix} \frac{\partial \ln \mathcal{N}(\boldsymbol{x}_1; \boldsymbol{\varphi}_1)}{\partial \boldsymbol{\varphi}_1} \frac{\partial \ln \mathcal{N}(\boldsymbol{x}_1; \boldsymbol{\varphi}_1)}{\partial \boldsymbol{\varphi}_1}^T, & \cdots, & \frac{\partial \ln \mathcal{N}(\boldsymbol{x}_1; \boldsymbol{\varphi}_1)}{\partial \boldsymbol{\varphi}_1} \frac{\partial \ln \mathcal{N}(\boldsymbol{x}_p; \boldsymbol{\varphi}_p)}{\partial \boldsymbol{\varphi}_p}^T \\ \vdots, & \ddots, & \vdots \\ \frac{\partial \ln \mathcal{N}(\boldsymbol{x}_p; \boldsymbol{\varphi}_p)}{\partial \boldsymbol{\varphi}_p} \frac{\partial \ln \mathcal{N}(\boldsymbol{x}_1; \boldsymbol{\varphi}_1)}{\partial \boldsymbol{\varphi}_1}^T, & \cdots, & \frac{\partial \ln \mathcal{N}(\boldsymbol{x}_p; \boldsymbol{\varphi}_p)}{\partial \boldsymbol{\varphi}_p} \frac{\partial \ln \mathcal{N}(\boldsymbol{x}_p; \boldsymbol{\varphi}_p)}{\partial \boldsymbol{\varphi}_p}^T \end{bmatrix}. \tag{19}$$

Denoted by $\boldsymbol{\mathcal{F}}_{ij} = \mathbb{E}_{\pi(\boldsymbol{x}; \boldsymbol{\varphi})}[\frac{\partial \ln \mathcal{N}(\boldsymbol{x}_i; \boldsymbol{\varphi}_i)}{\partial \boldsymbol{\varphi}_i} \frac{\partial \ln \mathcal{N}(\boldsymbol{x}_j; \boldsymbol{\varphi}_j)}{\partial \boldsymbol{\varphi}_j}^T]$ the block entry of $\boldsymbol{\mathcal{F}}(\boldsymbol{\varphi})$ on the $i$-th row and $j$-th column in (19), where $i \neq j$, then

$$\begin{aligned} \boldsymbol{\mathcal{F}}_{ij} &= \int_{\boldsymbol{x}} \pi(\boldsymbol{x}; \boldsymbol{\varphi}) \frac{\partial \ln \mathcal{N}(\boldsymbol{x}_i; \boldsymbol{\varphi}_i)}{\partial \boldsymbol{\varphi}_i} \frac{\partial \ln \mathcal{N}(\boldsymbol{x}_j; \boldsymbol{\varphi}_j)}{\partial \boldsymbol{\varphi}_j}^T d\boldsymbol{x} \\ &= \int_{\boldsymbol{x}} \prod_{k=1}^{p} \mathcal{N}(\boldsymbol{x}_k; \boldsymbol{\varphi}_k) \frac{\partial \ln \mathcal{N}(\boldsymbol{x}_i; \boldsymbol{\varphi}_i)}{\partial \boldsymbol{\varphi}_i} \frac{\partial \ln \mathcal{N}(\boldsymbol{x}_j; \boldsymbol{\varphi}_j)}{\partial \boldsymbol{\varphi}_j}^T d\boldsymbol{x} \\ &= \int_{\boldsymbol{x}} \prod_{k=1, k\neq i\neq j}^{p} \mathcal{N}(\boldsymbol{x}_k; \boldsymbol{\varphi}_k) \underbrace{\mathcal{N}(\boldsymbol{x}_i; \boldsymbol{\varphi}_i) \frac{\partial \ln \mathcal{N}(\boldsymbol{x}_i; \boldsymbol{\varphi}_i)}{\partial \boldsymbol{\varphi}_i}}_{=\frac{\partial \mathcal{N}(\boldsymbol{x}_i; \boldsymbol{\varphi}_i)}{\partial \boldsymbol{\varphi}_i}} \underbrace{\mathcal{N}(\boldsymbol{x}_j; \boldsymbol{\varphi}_j) \frac{\partial \ln \mathcal{N}(\boldsymbol{x}_j; \boldsymbol{\varphi}_j)}{\partial \boldsymbol{\varphi}_j}^T}_{=\frac{\partial \mathcal{N}(\boldsymbol{x}_j; \boldsymbol{\varphi}_j)}{\partial \boldsymbol{\varphi}_j}} d\boldsymbol{x} \\ &= \underbrace{\int_{\boldsymbol{x}_1} \cdots \int_{\boldsymbol{x}_p}}_{\int_{\boldsymbol{x}}} \prod_{k=1, k\neq i\neq j}^{p} \mathcal{N}(\boldsymbol{x}_k; \boldsymbol{\varphi}_k) \frac{\partial \mathcal{N}(\boldsymbol{x}_i; \boldsymbol{\varphi}_i)}{\partial \boldsymbol{\varphi}_i} \frac{\partial \mathcal{N}(\boldsymbol{x}_j; \boldsymbol{\varphi}_j)}{\partial \boldsymbol{\varphi}_j}^T \underbrace{d\boldsymbol{x}_1 \cdots d\boldsymbol{x}_p}_{d\boldsymbol{x}} \\ &= \prod_{k=1, k\neq i\neq j}^{p} \underbrace{\int_{\boldsymbol{x}_k} \mathcal{N}(\boldsymbol{x}_k; \boldsymbol{\varphi}_k) d\boldsymbol{x}_k}_{=1} \underbrace{\frac{\partial}{\partial \boldsymbol{\varphi}_i} \left( \overbrace{\int_{\boldsymbol{x}_i} \mathcal{N}(\boldsymbol{x}_i; \boldsymbol{\varphi}_i) d\boldsymbol{x}_i}^{=1} \right)}_{=\nabla 1 = \boldsymbol{0}} \underbrace{\frac{\partial}{\partial \boldsymbol{\varphi}_j} \left( \overbrace{\int_{\boldsymbol{x}_j} \mathcal{N}(\boldsymbol{x}_j; \boldsymbol{\varphi}_j) d\boldsymbol{x}_j}^{=1} \right)^T}_{=\nabla 1^T = \boldsymbol{0}^T} \\ &= \boldsymbol{0}, \end{aligned} \tag{20}$$

which means all the off-diagonal block entries are zero, and $\boldsymbol{\mathcal{F}}(\boldsymbol{\varphi})$ turns out to be a block-diagonal matrix. Furthermore, notice that $\boldsymbol{\mathcal{F}}(\boldsymbol{\varphi}_i) = \mathbb{E}_{\pi(\boldsymbol{x}; \boldsymbol{\varphi})}[\frac{\partial \ln \mathcal{N}(\boldsymbol{x}_i; \boldsymbol{\varphi}_i)}{\partial \boldsymbol{\varphi}_i} \frac{\partial \ln \mathcal{N}(\boldsymbol{x}_i; \boldsymbol{\varphi}_i)}{\partial \boldsymbol{\varphi}_i}^T]$ for all $i \in [p]$, which implies:

$$\begin{aligned} \tilde{\nabla} \ln \pi(\boldsymbol{x}; \boldsymbol{\varphi}) = \boldsymbol{\mathcal{F}}(\boldsymbol{\varphi})^{-1} \nabla \ln \pi(\boldsymbol{x}; \boldsymbol{\varphi}) &= \begin{pmatrix} \boldsymbol{\mathcal{F}}(\boldsymbol{\varphi}_1)^{-1}, & \boldsymbol{0}, & \boldsymbol{0} \\ \boldsymbol{0}, & \ddots, & \boldsymbol{0} \\ \boldsymbol{0}, & \boldsymbol{0}, & \boldsymbol{\mathcal{F}}(\boldsymbol{\varphi}_p)^{-1} \end{pmatrix} \nabla \ln \pi(\boldsymbol{x}; \boldsymbol{\varphi}) \\ &= \sum_{i=1}^{p} \boldsymbol{\mathcal{F}}(\boldsymbol{\varphi}_i)^{-1} \frac{\partial \ln \mathcal{N}(\boldsymbol{x}_i; \boldsymbol{\varphi}_i)}{\partial \boldsymbol{\varphi}_i}. \end{aligned} \tag{21}$$

As $\boldsymbol{x}$ follows $\pi(\cdot; \boldsymbol{\varphi})$, it is unbiased. $\qquad \square$

**Interpretion of (10).** We can plug (21) into (10) and apply Monte-Carlo approximation to $\tilde{\nabla} \bar{\mathcal{J}}_{\boldsymbol{\alpha}}(\boldsymbol{\theta})$, which yields:

$$\tilde{\nabla} \bar{\mathcal{J}}_{\boldsymbol{\alpha}}(\boldsymbol{\theta}) = \mathbb{E}_{\pi(\boldsymbol{x}; \boldsymbol{\varphi})} \left[ f(\boldsymbol{x}) \frac{\partial}{\partial \boldsymbol{\theta}} \left( \tilde{\nabla} \ln \pi(\boldsymbol{x}; \boldsymbol{\varphi}) \right) \right] = \frac{1}{\lambda} \sum_{j=1}^{\lambda} f(\boldsymbol{x}^j) \frac{\partial}{\partial \boldsymbol{\theta}} \left( \tilde{\nabla} \ln \pi(\boldsymbol{x}^j; \boldsymbol{\varphi}^j) \right), \tag{22}$$

where $\boldsymbol{\varphi}^j = (\boldsymbol{\varphi}_{1j}^T, ..., \boldsymbol{\varphi}_{pj}^T)^T$ and $\boldsymbol{\varphi}_{ij} = \boldsymbol{\varphi}_{\boldsymbol{\theta}}(\boldsymbol{x}_{0:i-1}^j)$. Intuitively, the right-hand side of the above works as follows: first, sample a bunch of distributions $\pi(\boldsymbol{\varphi}^1), ..., \pi(\boldsymbol{\varphi}^p)$; then, for each of these distributions, sample a

random solution $\boldsymbol{x}^j \sim \pi(\boldsymbol{\varphi}^j)$; in the meanwhile, $\boldsymbol{\theta}$ satisfies that $\boldsymbol{\varphi}_{ij} = \boldsymbol{\varphi}_{\boldsymbol{\theta}}(\boldsymbol{x}^j_{0:i-1})$; eventually, estimate the natural gradient of $\ln \pi(\boldsymbol{x}^j; \boldsymbol{\varphi}^j)$ w.r.t. $\boldsymbol{\varphi}^j$ using $\boldsymbol{x}^j$. In brief, *(10) amounts to aggregating the natural gradient of a set of sampled distributions $\pi(\boldsymbol{\varphi}^1), ..., \pi(\boldsymbol{\varphi}^p)$, and then back-propagating the gradient to $\boldsymbol{\theta}$ through the chain rule.*

## B.4  Deriving the update rule (11)

(22) cannot be directly used in implementing the algorithm, as it 1) involves computing $\mathcal{F}(\boldsymbol{\varphi}_i)^{-1}$, which is inefficient, and 2) does not specify how the gradient is back-propagated to $\boldsymbol{\theta}$ through the chain rule. To this end, we first introduce Equation (24) of Akimoto *et al.* [40], which avoids the computation of $\mathcal{F}(\boldsymbol{\varphi}_i)^{-1}$:

**Proposition 3** (Natural gradient of a Gaussian). *Consider a Gaussian random sample $\boldsymbol{x} \sim \mathcal{N}(\boldsymbol{\varphi})$ where $\boldsymbol{\varphi}_i = (\boldsymbol{m}_i^T, vec(\boldsymbol{C}_i)^T)^T$, the natural gradient of the log-likelihood $\ln \mathcal{N}(\boldsymbol{x}_i; \boldsymbol{\varphi}_i)$ is*

$$\mathcal{F}(\boldsymbol{\varphi}_i)^{-1} \frac{\partial \ln \mathcal{N}(\boldsymbol{x}_i; \boldsymbol{\varphi}_i)}{\partial \boldsymbol{\varphi}_i} = \begin{pmatrix} \boldsymbol{x}_i - \boldsymbol{m}_i \\ vec\left((\boldsymbol{x}_i - \boldsymbol{m}_i)(\boldsymbol{x} - \boldsymbol{m}_i)^T - \boldsymbol{C}_i\right) \end{pmatrix}. \tag{23}$$

Define $\boldsymbol{m}_i = \boldsymbol{m}_{\boldsymbol{\theta}}(\boldsymbol{x}_{0:i-1})$, $\boldsymbol{C}_i = \sigma_i^2(\boldsymbol{I} + \boldsymbol{v}_i \boldsymbol{v}_i^T)$, $\sigma_i = \sigma_{\boldsymbol{\theta}}(\boldsymbol{x}_{0:i-1})$, $\boldsymbol{v}_i = \boldsymbol{v}_{\boldsymbol{\theta}}(\boldsymbol{x}_{0:i-1})$, and $\boldsymbol{y}_i = \boldsymbol{x}_i - \boldsymbol{m}_i$. According to the chain rule:

$$\frac{\partial}{\partial \boldsymbol{\theta}} \left( \mathcal{F}(\boldsymbol{\varphi}_i)^{-1} \frac{\partial \ln \mathcal{N}(\boldsymbol{x}_i; \boldsymbol{\varphi}_i)}{\partial \boldsymbol{\varphi}_i} \right) = \overbrace{\boldsymbol{J}(\boldsymbol{m}_i)^T}^{=(\partial \boldsymbol{m}_i / \partial \boldsymbol{\theta})^T} (\boldsymbol{x}_i - \boldsymbol{m}_i) + \frac{\partial \operatorname{vec} \boldsymbol{C}_i}{\partial \boldsymbol{\theta}}^T \operatorname{vec}\left((\boldsymbol{x}_i - \boldsymbol{m}_i)(\boldsymbol{x} - \boldsymbol{m}_i)^T - \boldsymbol{C}_i\right)$$

$$= \boldsymbol{J}(\boldsymbol{m}_i)^T \boldsymbol{y}_i + \underbrace{\left( \frac{\partial \sigma_i}{\partial \boldsymbol{\theta}} \frac{\partial \operatorname{vec} \boldsymbol{C}_i}{\partial \sigma_i} + \boldsymbol{J}(\boldsymbol{v}_i)^T \frac{\partial \operatorname{vec} \boldsymbol{C}_i}{\partial \boldsymbol{v}_i} \right)}_{=(\partial \operatorname{vec} \boldsymbol{C}_i / \partial \boldsymbol{\theta})^T} \operatorname{vec}\left(\boldsymbol{y}_i \boldsymbol{y}_i^T - \boldsymbol{C}_i\right),$$

$$\tag{24}$$

where $\boldsymbol{J} : \mathbb{R}^d \to \mathbb{R}^{d \times |\boldsymbol{\theta}|}$ is the Jacobian matrix. In the following, we will expand the above terms one by one:

$$\frac{1}{2\sigma_i} \frac{\partial \operatorname{vec} \boldsymbol{C}_i}{\partial \sigma_i} \operatorname{vec}\left(\boldsymbol{y}_i \boldsymbol{y}_i^T - \boldsymbol{C}_i\right) = \operatorname{vec}(\boldsymbol{I} + \boldsymbol{v}_i \boldsymbol{v}_i)^T \cdot \operatorname{vec}(\boldsymbol{y}_i \boldsymbol{y}_i^T - \boldsymbol{C}_i)$$

$$= \operatorname{Tr}\left((\boldsymbol{I} + \boldsymbol{v}_i \boldsymbol{v}_i^T) \boldsymbol{y}_i \boldsymbol{y}_i^T\right) - \operatorname{Tr}\left((\boldsymbol{I} + \boldsymbol{v}_i \boldsymbol{v}_i^T) \overbrace{\sigma_i^2(\boldsymbol{I} + \boldsymbol{v}_i \boldsymbol{v}_i^T)}^{=\boldsymbol{C}_i}\right) \tag{25}$$

$$= \operatorname{Tr}\left(\boldsymbol{y}_i \boldsymbol{y}_i^T\right) + \operatorname{Tr}\left(\boldsymbol{v}_i \boldsymbol{v}_i^T \boldsymbol{y}_i \boldsymbol{y}_i^T\right) - \sigma_i^2 \operatorname{Tr}\left((\boldsymbol{I} + \boldsymbol{v}_i \boldsymbol{v}_i^T)^2\right)$$

$$= \|\boldsymbol{y}_i\|^2 + (\boldsymbol{v}_i^T \boldsymbol{y}_i)^2 - \sigma_i^2(d + 2\|\boldsymbol{v}_i\|^2 + \|\boldsymbol{v}_i\|^4);$$

and

$$\frac{\partial \operatorname{vec} \boldsymbol{C}_i}{\partial \boldsymbol{v}_i} \operatorname{vec}\left(\boldsymbol{y}_i \boldsymbol{y}_i^T - \boldsymbol{C}_i\right)$$

$$= \sigma_i^2 (\boldsymbol{I} \otimes \boldsymbol{v}_i + \boldsymbol{v}_i \otimes \boldsymbol{I})^T (\overbrace{\boldsymbol{y}_i \otimes \boldsymbol{y}_i}^{=\operatorname{vec}(\boldsymbol{y}_i \boldsymbol{y}_i^T)} - \operatorname{vec} \boldsymbol{C}_i)$$

$$= \sigma_i^2 (\boldsymbol{I} \otimes \boldsymbol{v}_i + \boldsymbol{v}_i \otimes \boldsymbol{I}) (\boldsymbol{y}_i \otimes \boldsymbol{y}_i) - \sigma_i^2 (\boldsymbol{I} \otimes \boldsymbol{v}_i + \boldsymbol{v}_i \otimes \boldsymbol{I}) \overbrace{\sigma_i^2 (\operatorname{vec}(\boldsymbol{I}) + \boldsymbol{v}_i \otimes \boldsymbol{v}_i)}^{=\sigma_i^2 \operatorname{vec}(\boldsymbol{I} + \boldsymbol{v}_i \boldsymbol{v}_i^T) = \operatorname{vec}(\boldsymbol{C}_i)} \tag{26}$$

$$= \sigma_i^2 \left((\boldsymbol{v}_i^T \boldsymbol{y}_i) \boldsymbol{y}_i + \|\boldsymbol{y}_i\|^2 \boldsymbol{v}_i\right) - 2\sigma_i^4 \left(1 + \|\boldsymbol{v}_i\|^2\right) \boldsymbol{v},$$

where $\otimes$ denotes the Kronecker product [61]. Plugging (25) and (26) into (24) yields:

$$(24) = \boldsymbol{J}(\boldsymbol{m}_i)^T \boldsymbol{y}_i$$

$$+ 2\sigma_i \left(\|\boldsymbol{y}_i\|^2 + (\boldsymbol{v}_i^T \boldsymbol{y}_i)^2 - \sigma_i^2(d + 2\|\boldsymbol{v}_i\|^2 + \|\boldsymbol{v}_i\|^4)\right) \frac{\partial \sigma_i}{\partial \boldsymbol{\theta}} \tag{27}$$

$$+ \sigma_i^2 \boldsymbol{J}(\boldsymbol{v}_i)^T \left(\boldsymbol{v}_i^T \boldsymbol{y}_i \boldsymbol{y}_i + \|\boldsymbol{y}_i\|^2 \boldsymbol{v}_i - 2\sigma_i^2 \left(1 + \|\boldsymbol{v}_i\|^2\right) \boldsymbol{v}\right).$$

Consider a sample $\boldsymbol{x}^j$, define $\boldsymbol{m}_{ij} = \boldsymbol{m}_{\boldsymbol{\theta}}(\boldsymbol{x}^j_{0:i-1})$, $\sigma_{ij} = \sigma_{\boldsymbol{\theta}}(\boldsymbol{x}^j_{0:i-1})$, $\boldsymbol{v}_{ij} = \boldsymbol{v}_{\boldsymbol{\theta}}(\boldsymbol{x}^j_{0:i-1})$, and $\boldsymbol{y}_{ij} = \boldsymbol{x}^j_i - \boldsymbol{m}_{ij}$. Eventually, for each $j \in [\lambda]$ substituting (27) into (22) yields (11).

## B.5  More on the weight values

The rank-dependent weights are also known as the use of utility [26, 25] or fitness shaping [20]. The rationale behind the weights in this paper, i.e., $w_1 \geq ... \geq w_\lambda$ and $w_j = \ln(\lfloor \frac{\lambda}{2} + \frac{1}{2} \rfloor) - \ln j$, is that they decrease

sub-linearly with the ranks: solutions before the median will be assigned positive weights, while those after (including the median itself) will be assigned negative weights, and no one will be assigned a zero weight. Separating the weights into positive and negative values is equivalent to using a reward baseline in many practical reinforcement learning methods. Note that in most CMA-ES variants [27, 29, 56], the second half of $\{\boldsymbol{x}^j\}_{j=1}^p$ is discarded by default, especially when updating the Gaussian mean. This is because using inferior solutions implicitly imposes too strong a regularity assumption on a black-box objective, which may not be true. A counterexample is xNES, which uses the inferior solutions but normalizes their weights to $-1/\lambda$. However, we adopt all solutions to estimate due to the well-known data-hungry nature of neural networks.

## B.6 Design of the set model

We use a feed-forward neural network as the set model. There are four hidden layers, each with a size of 1024. In order to input $\boldsymbol{x}_{0:i-1}$ into the model, we construct

$$
\Big( \overbrace{\boldsymbol{x}_1^T, \boldsymbol{x}_2^T, ..., \boldsymbol{x}_{i-1}^T,}^{=\boldsymbol{x}_{0:i-1}^T} \overbrace{\mathrm{nil}, ..., \mathrm{nil}}^{(p-i+1)d \ \mathrm{nil}\,s}, \overbrace{1, ..., 1}^{(i-1)d \ 1s}, \overbrace{0, ..., 0}^{(p-i+1)d \ 0s}, \boldsymbol{x}_0 \Big)^T,
\tag{28}
$$

where nil is a placeholder, 1 on the $(n+j)$-th $(1 \le j \le n)$ dimension indicates that the $j$-th input is not nil, and vice versa. Consequently, the size of the input layer is $2n + M$. The output layer, which outputs $(\boldsymbol{m}_i^T, \sigma_i, \boldsymbol{v}_i^T)^T$, is of size $2d + 1$. We use the leaky ReLU function as the activation in the hidden layers, which is defined as $f^{\mathrm{lr}}(\omega) = \max(\omega, 0) + 0.01 \min(0, \omega)$. In order to generate $\boldsymbol{m}_i, \boldsymbol{v}_i \in \mathbb{R}^d$, no activation is used in the output layer. To guarantee $\sigma_i > 0$, we take the absolute value of the $(d+1)$-th output dimension. The set model is trained using Adam, with a learning rate of $\eta = 10^{-3}$. Compared to SGD (stochastic gradient ascent), the adaptive momentum of Adam accumulates historical gradient information, enabling more accurate gradient estimation and significantly reducing the number of samples required.

## B.7 Overall computational complexity

Consider a set model, i.e., a feed-forward neural network, with $t$ hidden layers, and the sizes of the input, output, and hidden layers are $2n + M$, $2d + 1$, and $s$, respectively. One can derive the space complexity of the network, as well as the time complexities of the feed-forward and back-propagation processes, which are all $\Theta(ts^2 + (n + M + d)s)$. Since we are only interested in $n$ and $d$, we can further simplify the complexity as $\Theta(n + d)$. We can easily imply that the space complexity of Neural-ES is $\Theta(n + d)$ as well, as the entire search distribution is compressed into the model.

The time complexity of Neural-ES stems from two sources: 1) sampling and 2) updating $\boldsymbol{\theta}$. To sample a component $\boldsymbol{x}_i$, it takes $\Theta(n + d)$ time to generate a $\boldsymbol{\varphi}_i = \boldsymbol{\varphi}_{\boldsymbol{\theta}}(\boldsymbol{x}_{0:i-1})$, which corresponds to the feed-forward process, and $\Theta(d)$ time to generate an $\boldsymbol{x}_i$ according to (8). To generate a complete $\boldsymbol{x} = (\boldsymbol{x}_1^T, ..., \boldsymbol{x}_p^T)^T$, (8) needs to be performed $p$ times. Therefore, the overall time complexity for sampling a $\boldsymbol{x} \sim \pi$ is $p\Theta(n + d) + p\Theta(d) = \Theta(pn)$, as $n = pd$.

To update $\boldsymbol{\theta}$, we need to compute (27). Within the equation, the computation of $\boldsymbol{J}(\boldsymbol{m}_i)$, $\frac{\partial \sigma_i}{\partial \boldsymbol{\theta}}$ and $\boldsymbol{J}(\boldsymbol{v}_i)$ takes a total of $\Theta(n + d)$ time, which corresponds to the back-propagation process; every matrix-vector multiplication takes $\Theta(n + d) * \Theta(d) = \Theta(nd + d^2)$ time; in addition, the computation of $\|\boldsymbol{y}_i\|^2$, $\boldsymbol{v}_i^T \boldsymbol{y}_i$ and $\|\boldsymbol{v}_i\|^2$ takes $\Theta(d)$ time. To sum up, the time complexity of (27) is $\Theta(n + d) + \Theta(nd + d^2) + \Theta(d) = \Theta(nd + d^2)$. In addition, (27) needs to be performed $p$ times regarding a complete $\boldsymbol{x} = (\boldsymbol{x}_1^T, ..., \boldsymbol{x}_p^T)^T$, which takes $p\Theta(nd + d^2) = \Theta(n^2 + nd) = \Theta(n^2)$ time. Note that $\Theta(n^2)$ also represents the time complexity in the update of EPSL [15]. $\Theta(n^2)$ *is indeed a near-optimum one can expect when updating* $\boldsymbol{\theta}$, *because* $\frac{\partial \boldsymbol{\varphi}}{\partial \boldsymbol{\theta}}^T \frac{\partial \mathcal{J}}{\partial \boldsymbol{\varphi}}$ *achieves* $\Theta(n^2)$ *time only when both* $\boldsymbol{\varphi}$ *and* $\boldsymbol{\theta}$ *scale linearly with* $n$.

A typical counterexample is the application of other existing ES variants, such as xNES, to solve the black-box PSL problem. In this case, the space complexity will increase to $\Theta(n^2)$, the time complexity of sampling will increase to $\Theta(n^2)$, the time complexity of computing the partial derivative $\partial \bar{\mathcal{J}}_{\boldsymbol{\alpha}}(\boldsymbol{\theta})/\partial \boldsymbol{\varphi}$ will increase to $\Theta(n^2)$, and eventually computing the gradient $\tilde{\nabla} \bar{\mathcal{J}}_{\boldsymbol{\alpha}}(\boldsymbol{\theta})$ as well as updating $\boldsymbol{\theta}$ will cost $\Theta(n^3)$ time or higher. In contrast, *Neural-ES achieves* $\Theta(n)$ *space complexity,* $\Theta(pn)$ *time complexity of sampling a solution, and* $\Theta(n^2)$ *time complexity of updating* $\boldsymbol{\theta}$. *Therefore, we can conclude that Neural-ES is a time-efficient approach.*

# C More details on the experiments

## C.1 Proposed BBO-PSL benchmark suite

A test problem for assessing the PSL algorithms should satisfy the following two requirements:

- *Non-separability*, which means $\operatorname{argmin}_{\boldsymbol{x}} f_i(\boldsymbol{x}) \neq (\operatorname{argmin}_{x_1} f_i(x_1, ...), ..., \operatorname{argmin}_{x_n} f_i(..., x_n))^T$ for all $i \in [m]$. This is a *conditio sine qua non*, because separable problems, which are rarely the practical case, cannot assess the ability to handle complex dimension dependency.

- *Analytical PFs and complex PSs*: where the former means that the ground-truth PF and PS should be expressed in closed forms, and the latter suggests that the Pareto optimal solutions should not be, e.g., linearly distributed.

Surprisingly, few of the existing benchmark suites have satisfied both requirements. For example, the popular DTLZ problems [62] satisfy neither of them: they are separable, with simple PSs located on the boundary of the search space. Despite their complex PS shapes, the UF [44] and LZ [63] instances are separable. Though providing non-separable instances, the bi-objective BBOB problems [45] lack analytical PFs & PSs. The LS-MOP [46] focusing on large-scale multiobjective problems does not feature complex PSs, and most of its instances are separable or partially separable. An exception would be the recently proposed ZCAT suite [47], which, however, features non-simplex-like PFs and falls beyond the scope of this paper.

For these reasons, we design a novel benchmark suite based on the following theorem, which is also the design principle of the popular UF and LZ problems:

**Theorem 1** (Li & Zhang [63]). *Consider a MOP in (1) where*

$$f_i(\boldsymbol{x}) = f_i^{pf}(\boldsymbol{x}_I) + f^{dis}(\boldsymbol{x}_{II} - \boldsymbol{f}^{ps}(\boldsymbol{x}_I)), \tag{29}$$

$i = 1, ..., M$, $\boldsymbol{x}_I = (\mathrm{x}_1, ..., \mathrm{x}_{M-1})$, $\boldsymbol{x}_{II} = (\mathrm{x}_M, ..., \mathrm{x}_n)$, $\boldsymbol{x} \in \mathcal{X}$, $\mathcal{X} = \prod_{j=1}^{n}[a_j, b_j] \subset \mathbb{R}^n$, $f_i^{pf} : \prod_{j=1}^{M-1}[a_j, b_j] \to \mathbb{R}$, $\boldsymbol{f}^{ps} : \prod_{j=1}^{M}[a_j, b_j] \to \mathbb{R}_+^{n-M+1}$, and $f^{dis} : \prod_{j=1}^{n-M+1}[a_j, b_j] \to \mathbb{R}_+$. *Assume 1)* $f^{dis}$ *reaches the minimal of* $0$ *at* $\boldsymbol{0}$, *which means* $f^{dis}(\boldsymbol{x}) \geq f^{dis}(\boldsymbol{0}) = 0$, $\forall \boldsymbol{x} \in \mathbb{R}^n$, *and 2) the following MOP has a PF and PS, which are denoted by* $PF^*$ *and* $PS^*$,

$$\underset{\boldsymbol{x}_I \in \prod_{i=1}^{M-1}[a_i, b_i]}{\text{minimize}} (f_1^{pf}(\boldsymbol{x}_I), ..., f_M^{pf}(\boldsymbol{x}_I))^T. \tag{30}$$

*Then, 1) the PF of* $\boldsymbol{F}(\boldsymbol{x})$ *is* $PF^*$, *and 2) the PS of* $\boldsymbol{F}(\boldsymbol{x})$ *is* $\{(\boldsymbol{x}_I, \boldsymbol{x}_{II})^T | \boldsymbol{x}_{II} = \boldsymbol{f}^{ps}(\boldsymbol{x}_I), \boldsymbol{x}_I \in PS^*\}$.

Of the above, $f_i^{\text{pf}}$, $\boldsymbol{f}^{\text{ps}}$, and $f^{\text{dis}}$ are referred to as the PF, PS, and distance functions, as they determine the shapes of the PF and PS manifolds as well as the difficulty of optimizing $\boldsymbol{F}(\boldsymbol{x})$, respectively. With **Theorem 1**, analytical PFs and complex PSs can be implemented by designing proper $f_i^{\text{pf}}$ and $\boldsymbol{f}^{\text{ps}}$, and non-separability can be implemented by introducing non-separable functions that achieve the minimum of zero at $\boldsymbol{x} = \boldsymbol{0}$.

We design instances in the proposed benchmark suite as follows. All the instances employ the rotated ellipsoidal function as the distance function, which, considering an input $\boldsymbol{x}_{II}$, is expressed as:

$$f^{\text{rel}}(\boldsymbol{x}_{II}; \boldsymbol{\mathcal{R}}) = \boldsymbol{x}_{II}^T \boldsymbol{\mathcal{R}} \boldsymbol{\Lambda} \boldsymbol{\mathcal{R}}^T \boldsymbol{x}_{II}, \tag{31}$$

where $\boldsymbol{\mathcal{R}}$ is a $(n - M + 1)$-dimensional orthonormal matrix, and $\boldsymbol{\Lambda}$ is a $(n - M + 1)$-dimensional diagonal matrix whose $i$-th diagonal entry is $c^{\frac{i-1}{n-M}}$. Particularly, $c$ is called the condition number [53], which denotes the ratio between the largest and smallest eigenvalues of a Hessian, and typically controls the optimization difficulty; a larger $c$ yields a more difficult optimization problem. $\boldsymbol{\mathcal{R}}$ can be constructed by applying the Gram-Schmidt process to a random square matrix with standard normally distributed entries. All the bi-objective instances take the following form:

$$\boldsymbol{F}(\boldsymbol{x}) = \begin{cases} f_1^{\text{pf}}(x_1) + f^{\text{rel}}(\boldsymbol{x}_{II} - \boldsymbol{f}^{\text{ps}}(x_1); \boldsymbol{\mathcal{R}}) \\ f_2^{\text{pf}}(x_1) + f^{\text{rel}}(\boldsymbol{x}_{II} - \boldsymbol{f}^{\text{ps}}(x_1); \boldsymbol{\mathcal{R}}') \end{cases}, \tag{32}$$

where the PF function is $\boldsymbol{f}^{\text{pf}}(\boldsymbol{x_1}) = (x_1, 1 - \sqrt{x_1})^T$. All the tri-objective instances take the following form:

$$\boldsymbol{F}(\boldsymbol{x}) = \begin{cases} f_1^{\text{pf}}(x_1, x_2) + f^{\text{rel}}(\boldsymbol{x}_{II} - \boldsymbol{f}^{\text{ps}}(x_1, x_2); \boldsymbol{\mathcal{R}}) \\ f_2^{\text{pf}}(x_1, x_2) + f^{\text{rel}}(\boldsymbol{x}_{II} - \boldsymbol{f}^{\text{ps}}(x_1, x_2); \boldsymbol{\mathcal{R}}') \\ f_3^{\text{pf}}(x_1, x_2) + f^{\text{rel}}(\boldsymbol{x}_{II} - \boldsymbol{f}^{\text{ps}}(x_1, x_2); \boldsymbol{\mathcal{R}}'') \end{cases}, \text{ where } \begin{cases} f_1^{\text{pf}}(x_1, x_2) = \cos(0.5\pi x_1)\cos(0.5\pi x_2) \\ f_2^{\text{pf}}(x_1, x_2) = \cos(0.5\pi x_1)\sin(0.5\pi x_2) \\ f_3^{\text{pf}}(x_1, x_2) = \sin(0.5\pi x_1) \end{cases}, \tag{33}$$

and $\boldsymbol{\mathcal{R}}$, $\boldsymbol{\mathcal{R}}'$ and $\boldsymbol{\mathcal{R}}''$ denote three different rotation matrices.

Eventually, eight instances, F1 to F8, are listed in Table 2. F1 to F4 are bi-objective problems, while F5 to F8 are tri-objective. The search space for the bi-objective instances is $[0, 1] \times [-1, 1]^{n-1}$, and is $[0, 1]^2 \times [-2, 2]^{n-2}$ for the tri-objective ones. The condition number $c$ is 100 in F1 to F4, and is 2 in F5 to F8. Their PS manifolds in the $(x_1 - x_2 - x_3)$ subspace are displayed in Fig. 8. In short, the proposed suite features non-separable and anisotropic distance functions, covering diverse nonlinear PS manifolds.

Table 2: The proposed BBO-PSL benchmark suite.

| ID | $M$ | $\boldsymbol{f}^{\mathrm{ps}}$ |
|----|-----|--------------------------------|
| F1 | | $f_i^{\mathrm{ps}}(x_1) = 0.75(1-\psi_i)x_1 * \begin{cases} \cos(2\pi x_1), \text{ if } i \text{ is even,} \\ \sin(2\pi x_1), \text{ otherwise} \end{cases}$ |
| F2 | 2 | $f_i^{\mathrm{ps}}(x_1) = \begin{cases} (\psi_i - 1)(x_1^2 - 2x_1 + 0.5), \text{ if } i \text{ is even,} \\ x_1^{1+\psi_i} - 0.5, \text{ otherwise} \end{cases}$ |
| F3 | | $f_i^{\mathrm{ps}}(x_1) = (1-\psi_i)(2x_1 - 1)^3 + \psi_i(2x_1 - 1)$ |
| F4 | | $f_i^{\mathrm{ps}}(x_1) = 2x_1^{0.5+1.5\psi_i} - 1$ |
| F5 | | $f_i^{\mathrm{ps}}(x_1, x_2) = 0.5\xi_i \cos(\pi \frac{x_1+x_2}{2}) \cos(\pi|x_1 - x_2|)$ |
| F6 | | $f_i^{\mathrm{ps}}(x_1, x_2) = \xi_i \left(x_1^2(1-x_2) + (1-x_1^2)x_2 - \frac{1}{2}\right)$ |
| F7 | 3 | $f_i^{\mathrm{ps}}(x_1, x_2) = \xi_i(x_2(\sin(\pi x_1) - \frac{1}{2}) + (1-x_2)(\sin(-\pi x_1) + \frac{1}{2}))$ |
| F8 | | $f_i^{\mathrm{ps}}(x_1, x_2) = 2\xi_i(x_1 - x_2)(x_1 + x_2 - 1)$ |

$f_i^{\mathrm{ps}}(\cdot)$ denotes the $i$-th output of $\boldsymbol{f}^{\mathrm{ps}}$, $\psi_i = \frac{i-1}{n-2}$, and $\xi_i = \frac{n-i+1}{n-2}$.

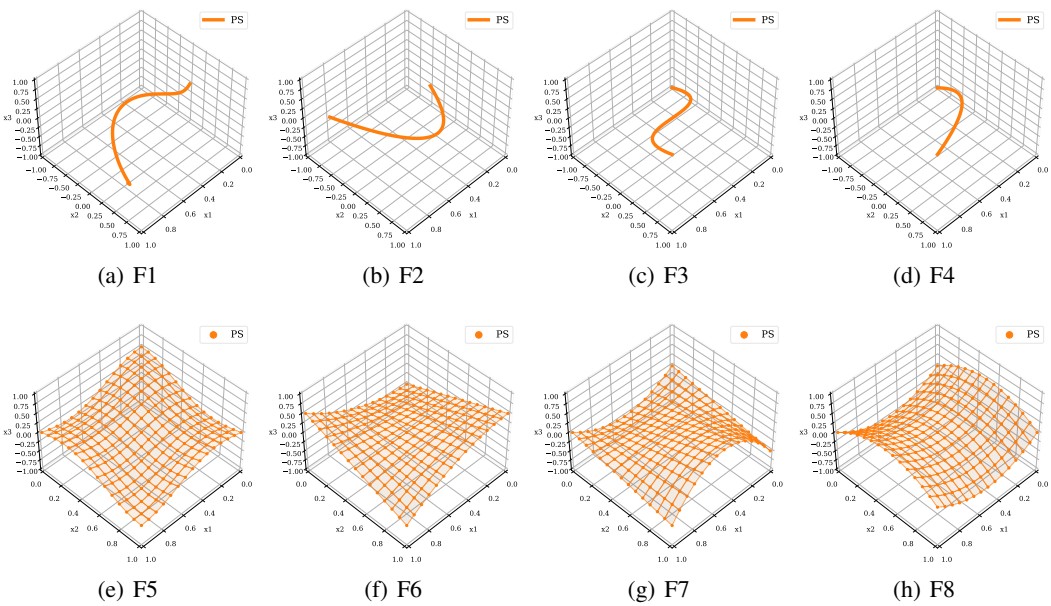

(a) F1    (b) F2    (c) F3    (d) F4

(e) F5    (f) F6    (g) F7    (h) F8

Figure 8: Ground-truth PS manifolds of the proposed problem instances.

## C.2 More about the configurations

**Compared algorithms.** The two MOEAs used in the experiments, namely LMO-CSO and IM-MOEA/D, are implemented in the open-source PlatEMO library [64] [4]. Their hyperparameters are configured to their default values. The set models of EPSL and EPSL-R1 use the same hidden-layer structure as Neural-ES. Bernoulli smoothing, antithetic sampling, and smooth Tchebycheff of EPSL are deactivated for a fair comparison. The most dominating hyperparameters of Neural-ES are $p$, $N$, and $\lambda$. While $\lambda = 4 + \lfloor 3\ln n \rfloor$ is a typical setup for most ES algorithms [19, 28], we reduce it to a half due to the large number of sampled subproblems $\{g(\cdot; \boldsymbol{\alpha}^k)\}_{k=1}^N$. We will discuss $p$ and $N$ in Section C.4. We use the popular Penalty-based Boundary Intersection (PBI) [8] as the aggregation function:

$$g_{\boldsymbol{u}}^{pbi}(\boldsymbol{F}; \boldsymbol{\alpha}) = D_1 + \rho D_2, \text{ where } \begin{cases} D_1 = \frac{1}{\|\boldsymbol{\alpha}\|}\|(\boldsymbol{u} - \boldsymbol{F})^T \boldsymbol{\alpha}\| \\ D_2 = \|\boldsymbol{F} - \boldsymbol{u} + D_1\boldsymbol{\alpha}\|, \end{cases}$$

$\boldsymbol{F}$ is a point in the objective space, $\boldsymbol{u} \in \mathbb{R}^M$ is a reference point satisfying $\boldsymbol{u} \leq (\inf f_1(\boldsymbol{x}), ..., \inf f_M(\boldsymbol{x}))^T$, $D_1$ is the distance from $\boldsymbol{u}$ to the projection of $\boldsymbol{F}$ onto $\boldsymbol{\alpha}$, and $D_2$ is the distance from $\boldsymbol{F}$ to $\boldsymbol{\alpha}$. More specifically, by minimizing $D_1$, one approaches the underlying PF, while minimizing $D_2$, which serves as a penalty, aligns

---

[4]https://github.com/BIMK/PlatEMO

with the preference $balpha$. The parameter $\rho > 0$ trades off $D_1$ and $D_2$. Although the Tchebycheff function [8] is also widely used, we have not considered it because we observe that its use can decrease solution diversity when solving tri-objective MOPs, which further leads to premature convergence of the set model.

**Performance metrics.**    The Inverted Generational Distance (IGD) [49] is used for assessing the algorithms' performance, with a smaller value indicating better performance. Denoted by $\widehat{PF}$ and $PF$ a subset of the ground-truth PF, and $PF$ an approximation by an algorithm, the IGD of $PF$ is defined as the total distance from every point in $PF$ to $\widehat{PF}$, and can be expressed as:

$$IGD(PF, \widehat{PF}) = \frac{1}{|\widehat{PF}|} \sum_{\boldsymbol{F}^* \in \widehat{PF}} \min_{\boldsymbol{F} \in PF} \|\boldsymbol{F} - \boldsymbol{F}^*\|_2.$$

Intuitively, IGD measures the distance from each point $\boldsymbol{F}^*$ in the ideal set $\widehat{PF}$, to the closest $\boldsymbol{F}$ in the approximation set $PF$. When imposing ideal properties on $\widehat{PF}$, the IGD is minimized only when $PF$ spreads widely and evenly. Otherwise, there will always be some $\boldsymbol{F}^* \in \widehat{PF}$ that remains distant from $PF$, resulting in a large IGD.

Additionally, hypervolume (HV) is also used to evaluate algorithms in Section 5.3 where IGD values are unavailable. It measures the volume of a specific region in the objective space, which is between a solution set and a reference point, with a larger value indicating better performance. It is given by:

$$HV_{\boldsymbol{u}}(S) = \text{Vol}\{\boldsymbol{y}|\exists \boldsymbol{x} \in S, \boldsymbol{F}(\boldsymbol{x}) \prec \boldsymbol{y} \prec \boldsymbol{u}\}$$

where $S$ is a solution set, $\boldsymbol{u}$ is a reference point in the $\boldsymbol{F}$-space, Vol computes the volume of a set, and $\prec$ denotes Pareto dominance. The volume always increases when the solution set approaches the ground truth PF more closely or spreads more widely across the PF, as both cases stretch the region.

**Experiment configurations.**    We limit the size of $\widehat{PF}$ to be 300 and 900 for the bi- and tri-objective instances, respectively. Each algorithm repeatedly solves each BBO-PSL instance 21 times, using a different random seed each time. For the three PSL algorithms, we evaluate their *testing performance*. Specifically, we input a large number of preferences into a trained set model, which yields the same number of distributions; from each distribution, we randomly sample a solution, the collection of which form the *testing solution set*; finally, we measure the IGD value of the set and refer it as the *testing IGD value*. The number of input preferences, i.e., the size of the testing solution set, is 900 and 4950 for the bi- and tri-objective instances, respectively. The MOEAs directly optimize the same number of solutions, and the IGD value of the final population is measured. Meanwhile, we track the convergence of the PSL algorithms during training, in terms of the IGD values of the *training solution set*. The training solution set at a certain iteration comprises the best solutions found so far for a set of evenly distributed subproblems. In light of the computational efficiency, its capacity is slightly smaller than the testing solution set, which is 100 and 300 for the bi- and tri-objective instances, respectively. To examine the scalability of the algorithms with increasing dimensions, we vary $n$ from 32 to 1024. The evaluation budget is $4000n$ and $2000n$ for bi- and tri-objective instances. The bi-objective instances are given a larger budget because their condition numbers are higher, which implies greater difficulties.

## C.3    More experiment results

**Numerical results.**    The numerical results for Fig. 4 are listed in Table 3. The Mann-Whitney U test at a $95\%$ confidence level is applied to statistically compare the algorithms. Neural-ES performs the best in all the instances, ranging from $n = 64$ to 1024. The results conform to those in Fig. 4. Note that the results of LMO-CMO and IM-MOEA/D are obtained through directly optimizing the problems, while the results of the three PSL algorithms represent their test-time generalization outcomes. This indicates that even the testing performance of Neural-ES surpasses that of the MOEAs in terms of optimization results.

**Approximate PF manifolds.**    Fig. 9 demonstrates the approximate PF manifolds by all five algorithms, on all the BBO-PSL instances. In all cases, the PFs approximated by Neural-ES dominate most areas of those approximated by the other algorithms, which conforms to Fig. 5. It appears that F3 and F4 are more challenging than F1 and F2, and F8 is more demanding than F5 to F7, indicating that the shape of the PS manifolds significantly influences the optimization difficulty. It can be observed that Neural-ES closely approximates the ground-truth PFs of all bi-objective instances, which highlights the learning capability of black-box PSL algorithms, particularly Neural-ES. However, the approximation results are inferior in the bi-objective instances, due to their significantly higher condition number. This confirms that the condition number of the distance function notably impacts optimization difficulty, which existing benchmark suites have neglected. Once again, the approximate PFs of Neural-ES and the other two PSL algorithms are produced during testing time, while those of LMO-CSO and IM-MOEA/D are direct optimization results. This suggests that the generalization ability of Neural-ES surpasses that of some state-of-the-art MOEAs.

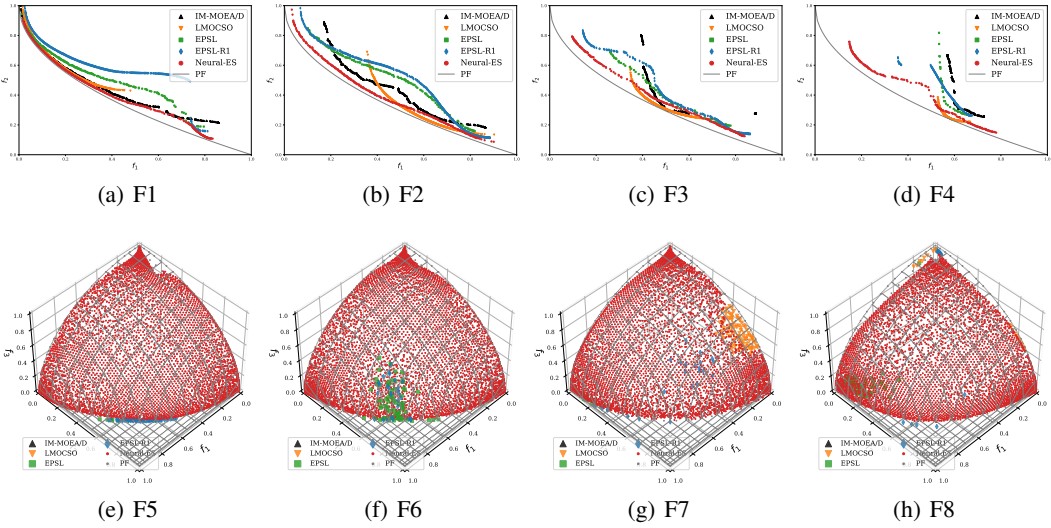

(a) F1         (b) F2         (c) F3         (d) F4

(e) F5         (f) F6         (g) F7         (h) F8

Figure 9: Testing-time PFs approximated by the algorithms, with $n = 256$. For better visualization, only non-dominated solutions are presented for F5 to F8.

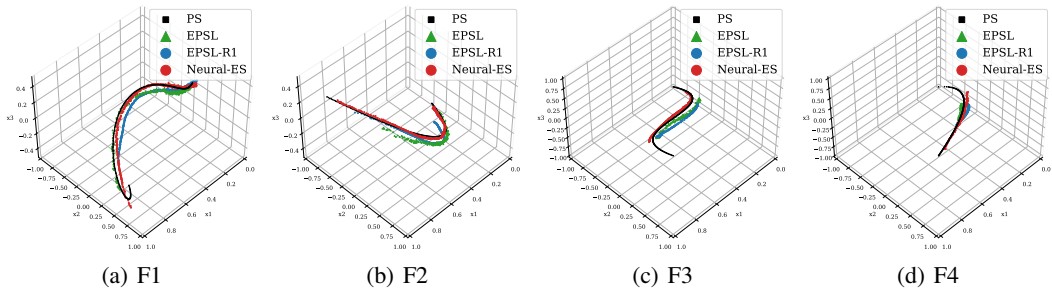

(a) F1         (b) F2         (c) F3         (d) F4

Figure 10: Testing-time PSs approximated by the PSL algorithms, on F1 to F4, with $n = 512$.

**Approximate PS manifolds.** We focus on the PSL algorithms' ability to learn the PS manifolds. LMO-CSO and IM-MOEA/D are excluded as they do not possess such ability. The approximate PSs on the bi-objective F1 to F4 are plotted in Fig. 10. Neural-ES generates the most accurate PSs in all four cases, as the red curves align the closest to the ground-truth. It can be observed that F3 and F4 are more challenging than F1 and F2, as they have higher levels of non-linearity. The approximate PSs on the tri-objective F5 to F8 are plotted in Figs. 11 to 14. Neural-ES and EPSL-R1 generate closer and more uniform approximations than EPSL in all four cases. Although Neural-ES only slightly outperforms EPSL-R1 in the solution space, their performance gap is significant, because most solutions of Neural-ES dominate those of EPSL-R1 in the objective space, as shown in Fig. 9.

**Convergence.** Fig. 15 demonstrates the convergence trajectories of the three PSL algorithms. LMO-CSO and IM-MOEA/D are not involved due to their differing sample sizes during each iteration. The results conform to Fig. 3. Noticeably, EPSL-R1 converges slightly faster than EPSL in F1 to F4; however, instances F5 to F8 do not reflect its advantage. This is because F1 to F4 have a much higher condition number of 100, than F5 to F8, which have a condition number of 2. Evidently, dealing with the dimensional dependencies becomes more challenging as the condition number increases. Once again, existing benchmark suites overlook this issue, necessitating the proposal of the BBO-PSL suite. In all the instances, Neural-ES establishes a performance lead over EPSL and EPSL-R1 during the early stages, extending the performance gap throughout the search process, underscoring its superiority in handling dimensional dependencies.

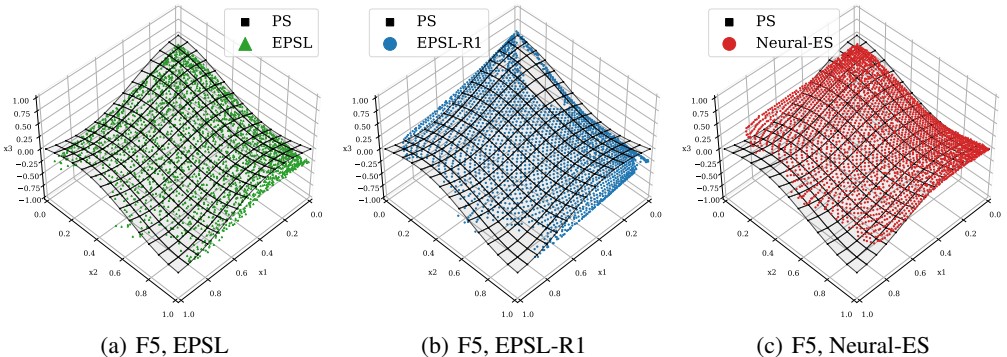

(a) F5, EPSL          (b) F5, EPSL-R1          (c) F5, Neural-ES

Figure 11: Testing-time PSs approximated by the PSL algorithms, on F5, with $n = 128$.

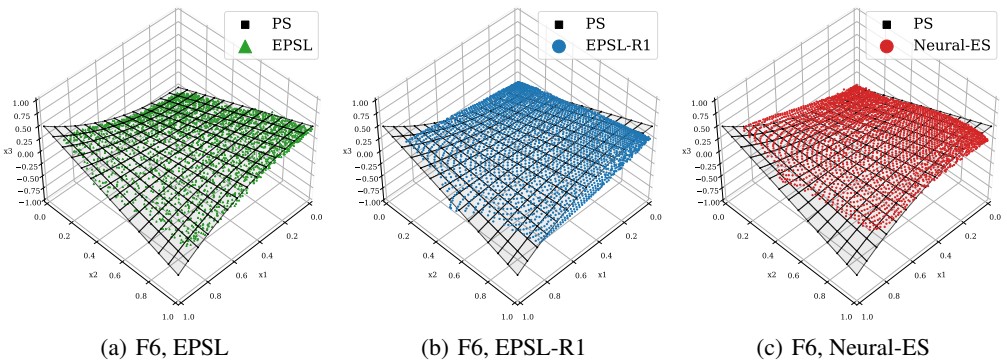

(a) F6, EPSL          (b) F6, EPSL-R1          (c) F6, Neural-ES

Figure 12: Testing-time PSs approximated by the PSL algorithms, on F6, with $n = 128$.

## C.4 Sensitivity analyses

Intuitively, the number of partitions, $p$, and the number of sampled subproblems, $N$, are closely linked to the performance of Neural-ES. To determine their optimal setups as well as to observe how they vary with the increasing problem dimension $n$, we conducted the following two independent experiments:

- *Investigating* $p$: we apply $p \in \{64, 32, 16, 8, 4\}$ to solve F4 and F6, and meanwhile, fix $N = \lfloor 5n/16 \rfloor$.

- *Investigating* $N$: we apply $N \in \{10, 20, 40, 80, 160, 320\}$ to solve F4 and F6, and meanwhile, fix $p = 16$.

In the experiments, the dimensions of F4 and F6 range from 64 to 512. The two selected instances serve as the representative bi- and tri-objective BBO-PSL instances, which are expected to provide the most neutral observations. The experiment results, in terms of the final IGD values, are recorded in Fig. 16.

We can make the following observations:

- Regarding Figs. 16(a) and 16(b), the number of partitions $p$ positively correlates to the problem dimension $n$. This is reasonable as a larger $p$ yields more decomposed Gaussians, and therefore a stronger ability to handle dimensional dependencies. The results validate the efficacy of the novel dependency handling technique presented in this paper (i.e., Equation (6)).

- Regarding Figs. 16(c) and 16(d), the highlighted main diagonals of the figures indicate the trade-offs between $N$ and Neural-ES: either too large or too small a sample size worsens the performance. This is reasonable because, on the one hand, a too large $N$ consumes a significant portion of the evaluation budget; on the other hand, a too small $N$ results in inaccurate gradient estimates.

To balance computational efficiency and performance, we adopt $p = 16$. It can be observed from Figs. 16(c) and 16(d) that the optimal $N$ is roughly $\lfloor 5n/16 \rfloor$. To date, we have justified the hyperparameter settings presented in this paper.

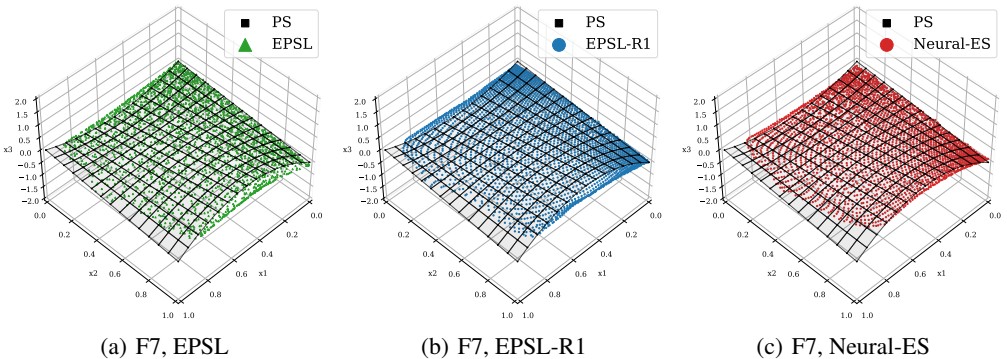

(a) F7, EPSL                     (b) F7, EPSL-R1                     (c) F7, Neural-ES

Figure 13: Testing-time PSs approximated by the PSL algorithms, on F7, with $n = 128$.

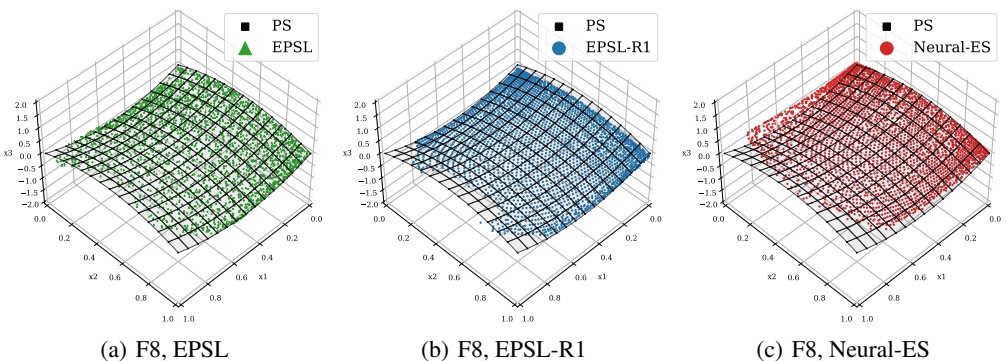

(a) F8, EPSL                     (b) F8, EPSL-R1                     (c) F8, Neural-ES

Figure 14: Testing-time PSs approximated by the PSL algorithms, on F8, with $n = 128$.

### C.5 Runtime analyses

We have conducted an extensive evaluation of computational efficiency across all compared algorithms using two 1024-dimensional instances (F1 and F5) from our BBO-PSL suite. The evaluation budget remained the same as before, at $4,000n$ for F1 and $2,000n$ for F5. The results are reported in Table 4.

Comparing the PSL algorithms with the traditional MOEAs, we can observe that the former run faster, or at least as fast, as the latter. This is because most MOEAs require advanced techniques to maintain population diversity and address slow convergence, such as the RVEA-based environmental selection in LMO-CSO and the Gaussian process modeling in IM-MOEA/D. Most, if not all, of these techniques come with high overhead and scale poorly with increasing population sizes. In contrast, the PSL paradigm does not rely on any of these techniques. Despite introducing neural networks, the PSL algorithms turn out to run surprisingly faster, or at least equally fast, when compared to traditional MOEAs.

Table 4: Wall-clock time (in minutes).

|    | LMO-CSO | IM-MOEA/D | EPSL | EPSL-R1 | Neural-ES |
|----|---------|-----------|------|---------|-----------|
| F1 | 16.9    | 17.4      | 7.8  | 12.2    | 17.4      |
| F5 | 23.9    | 15.2      | 5.1  | 7.4     | 9.5       |

While Neural-ES shows a marginally higher runtime compared to the other two PSL algorithms, due to its more complex search distribution, it achieves much faster convergence in terms of the number of generations required. The faster convergence likely offsets its marginally higher computational cost per generation, making it advantageous for practical applications.

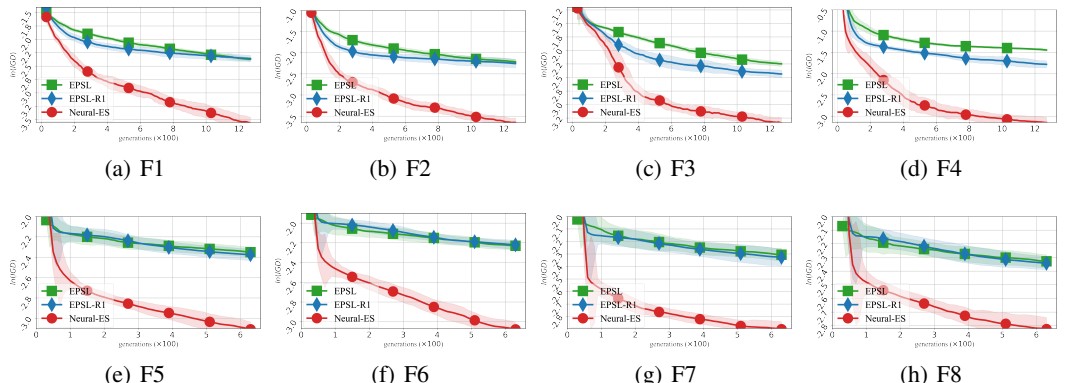

Figure 15: Convergence curves regarding the logarithm of the IGD values on the training solution sets, with $n = 256$.

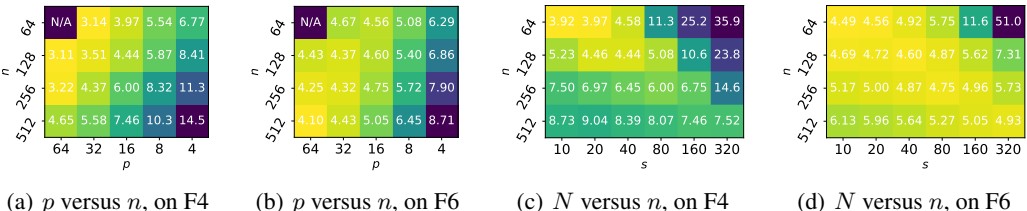

Figure 16: The acquired IGD values ($\times 10^{-2}$) in sensitivity analyses.

Table 3: Average training & testing IGD values ($\times 10^{-2}$) and standard deviations ($\times 10^{-3}$ in parentheses).

| | $n$ | LSO-CMO | IM-MOEA/D | EPSL | EPSL-R1 | Neural-ES |
|---|---|---|---|---|---|---|
| F1 | $2^5$ | 12.45 (47.98) − | **5.141** (16.66) | 7.202 (4.887) − | 6.946 (11.16) − | 6.099 (8.028) = |
| | $2^6$ | 15.77 (41.37) − | 5.570 (19.31) = | 7.187 (4.208) − | 8.216 (8.306) − | **4.797** (8.051) |
| | $2^7$ | 20.38 (30.18) − | 7.168 (21.02) − | 8.252 (4.229) − | 9.453 (8.892) − | **3.997** (5.794) |
| | $2^8$ | 22.11 (22.86) − | 7.231 (17.35) − | 9.758 (4.906) − | 12.10 (7.956) − | **3.967** (5.890) |
| | $2^9$ | 24.05 (9.820) − | 8.082 (12.90) − | 11.04 (3.594) − | 14.74 (6.150) − | **3.997** (4.171) |
| | $2^{10}$ | 26.62 (5.969) − | 10.69 (9.372) − | 12.71 (3.838) − | 16.88 (6.946) − | **5.336** (9.748) |
| F2 | $2^5$ | 4.993 (17.41) = | 6.526 (10.84) − | 7.016 (8.088) − | **4.588** (12.11) | 5.457 (16.46) = |
| | $2^6$ | 6.739 (19.71) − | 7.863 (14.81) − | 7.449 (7.552) − | 6.588 (15.80) − | **3.883** (11.72) |
| | $2^7$ | 8.445 (14.83) − | 8.726 (12.97) − | 8.773 (4.078) − | 7.581 (12.42) − | **3.004** (3.696) |
| | $2^8$ | 10.54 (12.55) − | 9.788 (10.07) − | 11.79 (7.447) − | 11.95 (17.83) − | **3.489** (5.859) |
| | $2^9$ | 15.13 (21.87) − | 13.20 (34.25) − | 13.59 (4.785) − | 15.13 (13.01) − | **3.727** (5.935) |
| | $2^{10}$ | 17.65 (16.18) − | 13.20 (15.28) − | 17.05 (6.887) − | 19.12 (12.04) − | **4.889** (5.704) |
| F3 | $2^5$ | 7.497 (12.54) − | 13.52 (9.551) − | 9.410 (7.694) − | **6.629** (7.717) | 6.692 (11.23) = |
| | $2^6$ | 9.351 (5.085) − | 14.30 (4.552) − | 9.486 (7.293) − | 7.477 (9.134) − | **5.771** (8.698) |
| | $2^7$ | 11.83 (5.836) − | 16.12 (4.709) − | 10.63 (5.437) − | 8.397 (7.823) − | **5.986** (9.637) |
| | $2^8$ | 14.24 (6.441) − | 16.61 (4.133) − | 11.68 (5.749) − | 10.13 (8.249) − | **5.863** (6.778) |
| | $2^9$ | 16.61 (8.691) − | 16.66 (2.910) − | 12.32 (5.782) − | 12.81 (7.739) − | **6.078** (4.259) |
| | $2^{10}$ | 19.13 (5.408) − | 16.76 (1.821) − | 14.22 (5.943) − | 15.81 (9.210) − | **6.654** (6.296) |
| F4 | $2^5$ | **4.647** (39.09) | 20.90 (13.96) − | 15.85 (18.37) − | 12.23 (13.27) − | 6.561 (12.06) = |
| | $2^6$ | 8.420 (28.84) − | 21.74 (11.57) − | 14.44 (14.62) − | 12.67 (18.09) − | **5.685** (9.287) |
| | $2^7$ | 17.70 (33.43) − | 23.77 (6.853) − | 21.53 (9.107) − | 15.71 (23.61) − | **6.558** (10.62) |
| | $2^8$ | 23.87 (15.02) − | 24.73 (6.528) − | 23.66 (7.163) − | 19.13 (12.07) − | **7.275** (8.371) |
| | $2^9$ | 27.69 (7.691) − | 25.25 (4.206) − | 24.64 (4.299) − | 22.88 (15.45) − | **8.728** (8.730) |
| | $2^{10}$ | 29.15 (2.736) − | 25.42 (7.191) − | 26.29 (5.367) − | 26.65 (11.38) − | **10.65** (10.97) |
| F5 | $2^5$ | 4.190 (2.751) − | 12.31 (12.62) − | 7.233 (6.933) − | 5.494 (6.135) − | **3.261** (5.537) |
| | $2^6$ | 5.815 (4.487) − | 19.31 (15.53) − | 8.065 (9.451) − | 6.052 (6.118) − | **3.090** (5.485) |
| | $2^7$ | 8.869 (7.198) − | 24.87 (18.05) − | 7.806 (6.459) − | 7.234 (7.670) − | **2.793** (4.118) |
| | $2^8$ | 12.22 (17.98) − | 32.98 (27.00) − | 8.830 (6.493) − | 7.982 (6.154) − | **3.147** (3.708) |
| | $2^9$ | 14.39 (16.78) − | 36.49 (21.15) − | 8.976 (6.300) − | 9.016 (11.79) − | **3.370** (4.695) |
| | $2^{10}$ | 16.79 (21.62) − | 39.58 (25.23) − | 9.383 (6.579) − | 9.764 (7.775) − | **3.937** (4.777) |
| F6 | $2^5$ | 4.528 (2.817) − | 13.23 (13.22) − | 7.454 (10.10) − | 5.991 (6.928) − | **2.752** (3.492) |
| | $2^6$ | 6.351 (5.353) − | 20.27 (16.66) − | 8.500 (8.668) − | 7.368 (6.723) − | **2.764** (3.685) |
| | $2^7$ | 9.792 (13.22) − | 33.81 (129.79) − | 8.930 (5.396) − | 8.382 (8.728) − | **2.635** (4.445) |
| | $2^8$ | 14.04 (23.30) − | 32.74 (18.43) − | 9.503 (4.951) − | 9.297 (6.391) − | **2.917** (3.655) |
| | $2^9$ | 18.06 (24.40) − | 37.48 (18.25) − | 10.10 (5.686) − | 10.33 (6.855) − | **3.261** (3.124) |
| | $2^{10}$ | 20.92 (41.97) − | 39.94 (11.84) − | 10.32 (4.974) − | 11.31 (13.02) − | **4.041** (4.753) |
| F7 | $2^5$ | 4.633 (3.024) − | 13.38 (17.32) − | 7.998 (6.516) − | 5.918 (4.893) − | **4.026** (6.118) |
| | $2^6$ | 6.498 (5.846) − | 18.77 (16.75) − | 8.120 (7.429) − | 6.694 (4.773) − | **4.166** (3.573) |
| | $2^7$ | 9.592 (13.00) − | 28.54 (50.67) − | 8.581 (4.237) − | 7.147 (5.947) − | **4.213** (3.333) |
| | $2^8$ | 12.69 (17.88) − | 31.79 (17.96) − | 9.277 (7.330) − | 8.386 (7.677) − | **4.330** (2.266) |
| | $2^9$ | 17.31 (27.64) − | 34.64 (12.73) − | 9.585 (6.458) − | 9.084 (7.185) − | **4.523** (2.586) |
| | $2^{10}$ | 21.01 (23.59) − | 37.25 (17.04) − | 9.726 (5.192) − | 9.935 (8.296) − | **4.833** (1.760) |
| F8 | $2^5$ | **4.833** (3.210) | 12.41 (22.91) − | 7.790 (7.609) − | 6.726 (7.825) − | 5.105 (5.517) = |
| | $2^6$ | 6.765 (5.986) − | 18.51 (12.94) − | 8.413 (9.989) − | 6.936 (5.334) − | **4.472** (4.858) |
| | $2^7$ | 9.127 (8.242) − | 25.61 (22.68) − | 8.663 (7.097) − | 7.340 (5.793) − | **4.220** (3.594) |
| | $2^8$ | 12.44 (14.13) − | 30.38 (15.34) − | 9.471 (7.997) − | 8.243 (6.328) − | **4.347** (3.694) |
| | $2^9$ | 16.69 (22.96) − | 34.29 (16.27) − | 9.076 (5.806) − | 8.655 (5.327) − | **4.477** (2.702) |
| | $2^{10}$ | 19.06 (42.91) − | 36.74 (17.06) − | 9.462 (4.698) − | 9.316 (5.610) − | **4.939** (3.172) |

.

