# OpenReview forum: "Neural Evolution Strategy for Black-box Pareto Set Learning"
_NeurIPS.cc/2025/Conference — NeurIPS 2025 poster_

### Official Review · Reviewer_evfh · 2025-06-06

**Clarity:** 4
**Significance:** 3
**Originality:** 4
**Rating:** 4
**Confidence:** 5

**Summary:**

The author proposes a new method called Neural-ES, which integrates evolutionary strategies into the paradigm of Pareto set learning. The key idea is to use neural networks to model and capture the complex dimensional dependencies in high-dimensional design variables, rather than relying on the traditional evolution strategy covariance matrix.

**Questions:**

See Weaknesses part.

**Ethical Concerns:**

["NO or VERY MINOR ethics concerns only"]

**Final Justification:**

My concerns have been addressed, so I maintain my positive score.

**Limitations:**

Yes

**Paper Formatting Concerns:**

No formatting issues.

**Quality:**

4

**Strengths And Weaknesses:**

Strengths:
1. This paper proposed a novel method for solving the Black-box Pareto Set Learning problem.

2. The experimental setting is perfect, and a large number of numerical test results prove the effectiveness of the method.

Weaknesses:
1. I checked the provided three refs on Line 230. These papers do not seem to provide a clear convergence rate of the ES method. The author should provide more discussion on the theoretical convergence properties of this method. Can it directly link to the convergence analysis of the standard ES method? Can the authors add a discussion with "Black-box Optimizer with Implicit Natural Gradient" about the connection between methods and convergence?

2. I still have concerns about the Pareto properties of the solved MOO. It seems the original problem is solving $F(x)$. But the proposed method is finding the gradient of $J(\theta)$, is the solution of optimizing $J(\theta)$ the solution of minimizing $F(x)$. In the convex case, I think the ES method works, but in the non-convex case, the Pareto-stationay point of the distribution approximated objective and original objective seems not equal. (See "Adaptive Stochastic Gradient Algorithm for Black-box Multi-Objective Learning")

3. Is there any practical application scenario here, about Pareto Set Learning in the black box scenario? We have lots of Pareto optimization methods, such as PaMaL, LORPMAN, COSMOS, PHN, why do we need it (for the black-box case). In many numerical cases, I think the advantage of black-box optimization is it can solve non-smooth or non-continuous, or complex functions, but ES-based method seems do not have a theoretical guarantee on those problems.

---

> ### Author Rebuttal · Authors · 2025-07-31
>
> Thank you for your insightful and constructive comments and suggestions. We appreciate that you recognized our method as novel and commended our experimental setting. Below, please find our detailed responses.
>
> ------
>
> > ### **W1** Theoretical convergence rate
>
> **A1**: We appreciate the insightful and constructive comments and break down our response into the following aspects:
>
> __Analyses on the convergence rate of ES remain challenging__. Recently have seen a significant development towards the convergence rate of ES methods, where some empirical and theoretical evidence suggests that ES converges linearly on some specific functions. For example, both Glasmachers [1] and Akimoto et al. [2] utilized the drift analysis to prove the convergence, where an overall Lyapunov function of the state of the algorithm (distribution mean and step-size) is used to prove upper and lower bounds of the expected first hitting time of an epsilon neighborhood of the optimum. However, most of the existing works heavily rely on fierce assumptions, which can be unfounded in practice. For example, the analyses of Glasmachers and Akimoto are limited to the simple (1+1)-ES, and the scale of the covariance matrix in Akimoto’s method needs to be strictly bounded.
>
> __Convergence analysis for multiobjective optimization algorithms is limited__. In multiobjective optimization, seeking a solution set rather than a single Pareto optimal solution makes the analysis even more challenging. This is because the most commonly used Pareto dominance is a partial order rather than a total order, making it difficult to define the optimality for a set-based optimization algorithm. Although analyzing the convergence to a single Pareto stationary point remains possible, which has been elegantly pursued in works like [3], there has been limited research, to the best of our knowledge, investigating the theoretical convergence under Pareto dominance.
>
> __Discussions with INGO__: In [4], Lyu and Tsang proposed a black-box optimization algorithm called Implicit Natural Gradient Optimization (INGO). To avoid explicitly inverting the Fisher, they formulated a surrogate objective where the KL-divergence is attached as a penalty term and can be explicitly derived (Theorem 3). The authors showed that the gradient with respect to the expectation of the sufficient statistics amounts to the natural gradient of the natural parameter of an exponential-family distribution. Furthermore, the authors proved that INGO achieves sublinear convergence when the surrogate objective is $\gamma$-strong convex (Theorem 4). For Lipschitz continuous, convex, and black-box functions, the authors proved that INGO also achieves sublinear convergence, the overhead of which can be further improved when applying orthogonal sampling (Theorem 5). We compare Neural-ES with INGO as follows:
> * Both Neural-ES and INGO avoid explicitly inverting the Fisher, which substantially reduces the variance of the estimated gradient and increases the computational efficiency.
> * Both the empirical evidence in Fig. 15 of the supplementary document and the theoretical results of INGO suggest that both algorithms achieve sublinear convergence.
> * It may be feasible to replace Eq. (10) with the implicit natural gradient formulation of INGO, but the subsequent deviations need to be changed accordingly.
> * However, as mentioned above, the primary difficulty of convergence analysis with the PSL paradigm arises from the lack of a definition of optimality based on a total order. As a result, the convergence analysis used in INGO may not be easily applicable to Neural-ES.
>
> To summarize, recent findings show that some ES algorithms achieve linear global convergence, but strong assumptions are required; meanwhile, the difficulty of obtaining the convergence rate for set-based multiobjective optimization algorithms (and PSL as well) mostly arises from the lack of a clear definition of total-order optimality. These two main reasons obstruct the analysis for the proposed Neural-ES. Nonetheless, we believe your insightful suggestion clearly helps improve the quality of our work, and will add the above discussions to the revised manuscript.
>
> ------
>
> ### **W2:** Convergence of the surrogate $\mathcal{J}(\theta)$
>
> **A2:** In [3], Ye et al. proposed the Adaptive Stochastic Multi-objective Gradient (ASMG) algorithm for black-box MOPs, extending the INGO algorithm with promising results. A key finding is that, in black-box scenarios, ASMG may only converge to an $\epsilon$-accurate Pareto stationary point rather than an exact one (Propositions 4.5, 4.6). From our understanding, this is because the Gaussian smoothing relaxes the original KKT conditions in MOPs, which means the gradient norm of the original $F$ can still be slightly greater than zero, but is still strictly less than the upper bound set by the covariance norm, when the surrogate $\mathcal{J}$ achieves Pareto stationary. We fully agree with this observation as it may be a common issue for most black-box MOO algorithms, including Neural-ES. We appreciate you bringing it to our attention, and will clarify this point in detail (e.g., in both Sec. 3.1 and the limitation) in the revised manuscript.
>
> ------
>
> > ### **W3**: Practical applications
>
> **A3**: Fully acknowledging the necessity for more practical applications, we have supplemented our work with new experiments on a tri-objective unmanned aerial vehicle (MO-UAV) navigation problem. The problem is included in the Meta-BBO-v2 library [5,6]. We have examined the PSL algorithms on three problem instances with varying difficulty levels. The primary results are as follows:
>
> | Instance | Dimensions | Neural-ES (HV) | EPSL (HV) | EPSL-R1 (HV) |
> |:---------:|:----------:|:--------------:|:------------:|:-------------:|
> | Easy | 30 | **3.397±0.279**  | 2.471±0.356 | 1.577±0.149 |
> | Moderate | 60 | **3.443±0.452**  | 2.399±0.372 | 1.520±0.267 |
> | Hard | 120 | **2.442±1.143**  | 0.716±0.197 | 0.842±0.139 |
>
> The results highlight the efficiency of Neural-ES. On one hand, PSL is capable of most categories of black-box MOPs as most other algorithms (e.g., MOEAS) do; on the other hand, its merit lies in the ability to "provide a tailored optimal solution for any valid decision-maker preference trade-off without requiring re-optimization from scratch", as we stated in lines 31-32 of the main text.
>
> Besides, we totally agree with you that the "advantage of black-box optimization is it can solve non-smooth or non-continuous, or complex functions". In fact, some recent studies [7,8] have proven that ES converges globally and linearly, on more than convex functions, such as quasiconvex functions and discontinuous functions, as well as those with non-convex sublevels. Therefore, combining ES with PSL could be one of the many promising directions for tackling black-box MOPs.
>
> ------
>
> > ### **References**
>
> [1] T. Glasmachers. "Global convergence of the (1+1) evolution strategy to a critical point." Evolutionary computation 28.1 (2020): 27-53.
>
> [2] Y. Akimoto, et al. "Global linear convergence of evolution strategies on more than smooth strongly convex functions." SIAM Journal on Optimization 32.2 (2022): 1402-1429.
>
> [3] F. Ye, et al. "Adaptive stochastic gradient algorithm for black-box multi-objective learning." ICLR. 2024.
>
> [4] Y. Lyu, and I.W. Tsang. "Black-box optimizer with stochastic implicit natural gradient." ECML-PKDD, 2021.
>
> [5] Z. Ma, et al. "Metabox: A benchmark platform for meta-black-box optimization with reinforcement learning." NeurIPS 36 (2023): 10775-10795.
>
> [6] Z. Ma, et al. "MetaBox-v2: A Unified Benchmark Platform for Meta-Black-Box Optimization." arXiv preprint arXiv:2505.17745 (2025).
>
> [7] A. Auger, and N. Hansen. "Linear convergence of comparison-based step-size adaptive randomized search via stability of Markov chains." SIAM Journal on Optimization 26.3 (2016): 1589-1624.
>
> [8] C. Toure, A. Auger, and N. Hansen. "Global linear convergence of evolution strategies with recombination on scaling-invariant functions." Journal of Global Optimization 86.1 (2023): 163-203.

---

> > ### Comment · Reviewer_evfh · 2025-08-03
> >
> > Thanks for the authors' responses. From a mathematical perspective, after reformulating the original problem using Eq. 4 (regardless of whether the parametric distribution is a Gaussian distribution), we should discuss whether the two optimization problems are equivalent, thereby demonstrating the reliability of optimizing the reformulated problem. Based on the authors' response, the current version of the paper lacks such a discussion. My concerns have been addressed, and I believe this paper meets the acceptance bar of NeurIPS, so I maintain my positive score.

---

> > > ### Author Response · Authors · 2025-08-04
> > > **Thank you**
> > >
> > > Thank you very much for your time, effort, and active participation in reviewing our work. Your invaluable feedback significantly enhances the quality of our manuscript, and we will revise it accordingly and provide more in-depth discussions on Eq. (4).

---

### Official Review · Reviewer_W93V · 2025-07-01

**Clarity:** 3
**Significance:** 2
**Originality:** 2
**Rating:** 4
**Confidence:** 2

**Summary:**

This paper proposes Neural-ES, a black-box optimization algorithm for Pareto set learning in high-dimensional multi-objective problems. The method replaces explicit full covariance matrices with a neural network that outputs a low-rank covariance structure in an autoregressive sampling framework. The authors demonstrate the algorithm on the proposed BBO-PSL benchmark suite and a real-world trajectory planning problem, showing competitive or superior performance compared to state-of-the-art MOEAs.

**Questions:**

1. Could the authors clarify why the results for EPSL-R1 are not reported in Figure 6(d)?

2. Could the authors provide more details on how the BBO-PSL benchmark suite was designed and constructed?

**Ethical Concerns:**

["NO or VERY MINOR ethics concerns only"]

**Limitations:**

Yes

**Quality:**

2

**Strengths And Weaknesses:**

Strengths:

1. The paper clearly articulates why existing methods fail to scale in high-dimensional multi-objective settings. The problem is well-motivated and practically relevant.

2. The mathematical derivations, including the gradient estimation and the natural gradient correction, are rigorous and clearly presented.

3. The authors designed the BBO-PSL benchmark, which covers a diverse set of challenging properties (nonseparability, scalability, known Pareto sets).


Weaknesses:

 1. The hyperparameters are fixed across experiments, without a sensitivity analysis.

2. The paper does not include sufficient ablation experiments. This makes it harder to attribute performance gains to specific components.

3. No runtime measurements are reported. It is unclear how much additional time the autoregressive sampling introduces compared to standard ES variants. This makes it hard to assess the method’s practical efficiency.

 4. Most algorithmic components (low-rank approximation, natural gradient) are adaptations of prior work. The main novelty is in combining autoregressive sampling with ES.

---

> ### Author Rebuttal · Authors · 2025-07-30
>
> We sincerely appreciate the reviewer's thorough evaluation and constructive feedback. Below, we provide point-by-point responses to address all raised concerns with a clear distinction between:
> * Existing content (simple past tense)
> * New analyses (present perfect tense)
>
> ------
>
> > ### **W1:** Hyperparameter sensitivity analysis
>
> **A1:** As detailed in Section C.4 of the supplementary material, we conducted sensitivity analyses on two critical hyperparameters, which are:
> * $p$, the partition count, and
> * $N$, the number of sampled subproblems.
>
> As mentioned, the results in Fig. 16 indicated that
> * A larger $p$ usually leads to better performance, because it increases the number of Gaussian components and thus enhances the ability to decouple dimensional dependencies;
> * Either too large or too small of $N$ deteriorates the performance, because the former consumes too much evaluation budget while the latter reduces the precision of the estimated gradient.
>
> Throughout the paper, we set $p=n/16$ as it balances performance and computational efficiency, and we set $N=5n/16$, which yields the best performance in general, according to Figs. 16(c) and (d).
>
> ------
>
> > ### **W2:** Ablation study
>
> **A2:** Neural-ES is designed based on a minimalistic principle, containing few modular components. As illustrated in Fig.1, Neural-ES primarily consists of two parts: the set model and the back-end search distribution. Our core contribution is a novel formulation of the search distribution, which endows PSL with the ability to effectively deal with dimensional dependencies.
>
> The experiments on EPSL and EPSL-R1 in Sec. 5 serve dual purposes: they are both comparisons between Neural-ES and the state-of-the-art methods and ablation studies. More specifically,
> * The comparison versus EPSL-R1, which is synthesized by “setting p = 1 in Neural-ES”, and “represents the simplest way to integrate PSL with an ES” (lines 276~277), validates the proposed search distribution;
> * The comparison versus EPSL, which is “a finite difference approach is employed to train the set model” (line 113), validates the ability to handle dimensional dependencies.
>
> ------
>
> > ### **W3:** Runtime benchmarking
>
> **A3:** Acknowledging the necessity for runtime assessment, we have supplemented new experiments that will be included in the revised manuscript. We have measured wall-clock time on the 1024-dimensional F1/F5 instances of our proposed BBO-PSL suite. The evaluation budget is 4000n for F1, and 2000n for F5. The results are reported as follows:
> | Algorithm | F1 | F5 |
> |:------------:|:---------:|:---------:|
> | LMO-CSO | 1019s (16.9min) | 1432s (23.9min) |
> | IM-MOEA/D | 1044s (17.4min) | 911s (15.2min) |
> | EPSL | 486s (7.76min)  | 308s (5.14min) |
> | EPSL-R1 | 731s (12.2min)  | 443s (7.39min) |
> | Neural-ES | 1042s (17.4min) | 567s (9.45min) |
>
> Comparing the PSL algorithms with the traditional MOEAs, we can observe that the former run faster, or at least as fast, as the latter. This is because most MOEAs rely on additional techniques to maintain population diversity and to deal with slow convergence. Most, if not all, of these techniques come with high overhead and scale poorly with increasing population sizes. In contrast, the PSL paradigm does not rely on any of these techniques, and it turns out that training the set model is less costly than usually believed when compared to MOEAs.
>
> Comparing Neural-ES to EPSL and EPSL-R1, the former runs slightly slower than the latter two, but remains competitive overall. As mentioned in Sec.B7 of the supplementary document, Neural-ES yields $\mathcal{O}(n)$ time complexity when ignoring the network inference cost and $\mathcal{O}(n^2)$ when including it. To compare, EPSL and EPSL-R1 exhibit the same complexities, only that they have lower overhead due to simpler search distributions.
>
> ------
>
> > ### **Q1:** Display issue
>
> **A4:** In Fig. 6(d), solutions generated by EPSL-R1 were unreported because they were dominated by solutions of Neural-ES and EPSL. As noted in the caption of Fig.6, we only displayed solutions that were not dominated by others for clarity. Similar situations occurred in Figs. 9(e) to (h) of the supplementary document, where, at times, the results for EPSL-R1 were not displayed, while at other times, it could be the other algorithms that were missing.
>
> ------
>
> > ### **Q2:** Details of the BBO-PSL suite
>
> **A5:** The details of the BBO-PSL suite were presented in Sec. C.1 of the supplementary document. In brief, the instances in the suite were designed based on Theorem 1 in the supplementary document. Three functions $f^{pf}$, $f^{ps}$, and $f^{dis}$ independently control the shapes of the PF and PS, as well as the optimization difficulty. Notably, the distance function $f^{dis}$ injects complex dimensional dependencies into the problems by utilizing the popular rotated ellipsoidal function (i.e., Eq. (31)). We depict the ground-truth PSs in the ($x_1-x_2-x_3$) subspace in Fig. 8, which shows their diverse and highly non-linear structures.

---

### Official Review · Reviewer_37Ck · 2025-07-03

**Clarity:** 4
**Significance:** 3
**Originality:** 4
**Rating:** 5
**Confidence:** 4

**Summary:**

The paper proposes a new method for black-box multiobjective optimization based on Pareto Set Learning (PSL). The approach relies on a Evolution Strategy (ES) scheme to conduct the optimization for specific objectives preference (trade-off), which search for specific preference is fed back to PSL for updating the general PSL set model. The PSL set model provides the sampling distribution used by the ES for the given preference. Results show significantly better performances that several other multi-objective optimization techniques applied to black box problem, including approaches based on PSL.

**Questions:**

- Can you provide a more detailled assessment of the overall efficiency of the proposed Neural-ES approach regarding the number of samples to evaluate for converging to good set compared to other approaches?
- I think there is a notion of coverage over the Pareto front is important for the current problem, but this is not much explored, metrics used are IGD and hypervolume. Can you explain how these metrics relate to the fact that the Pareto front is well covered (or not) from a given optimization run?

**Ethical Concerns:**

["NO or VERY MINOR ethics concerns only"]

**Final Justification:**

I would like to thanks the authors for their answers and extra experiments conducted. If not already done, I am advising strongly to update the paper according to the discussions with the reviewers. I am satisfied with the answers and I am maintaining my evaluation.

**Limitations:**

There is not much direct impact on society coming from the current work, so no discussion is required here. The Limitation paragraph in the Conclusion (sec. 6) consider constraint in multi-objective optimization as a future work is quite out-of-scope and seems to be a general problem with PSL. I am not sure it is much relevant to mention this in the paper.

**Paper Formatting Concerns:**

Looks good, no formatting concerns noticed.

**Quality:**

3

**Strengths And Weaknesses:**

Strengths:
- A quite straightforward and sound proposal to extend PSL to black box multi-objective optimization.
- Detailed presentation of the method, with enough details that should allow to reproduce the method from the paper rather easily.
- The paper is quite well-written and easy to follow.

Weaknesses:
- Black box optimization usually requires more potential solutions (samples) to be evaluated for achieving good enough solutions compared to cases where analytical equations of the objectives are available. Except for Fig. 3, which shows convergence over generations in a specific case, this is not considered in much details in the paper, while being an important aspect to consider, in my opinion.
- The experiments are not always easy to follow and the figures presented are not that obvious to understand. For instance, from Fig. 6 (a)-(c), I am not sure about conclusions to have.
- Likewise, the practical application in Sec. 5.3 is not super convincing, the results in Fig. 7 showing simply that the proposed Neural-ES approach is skewed toward low f2 values, while EPSL-R1 is the opposite. Many of the results presentated appear more visual/qualitative than anything.

---

> ### Author Rebuttal · Authors · 2025-07-30
>
> Thank you very much for your supportive recommendation and insightful comments. We are thrilled that you found our work straightforward, sound, and easy to follow. We will include all the suggested details and explanations in the revised manuscript. Below, please see our detailed responses.
>
> ------
>
> > ### **W1: & Q1**: Number of samples required for convergence.
>
> **A1:** The total number of samples is proportional to the total number of evaluations. In each generation of the Neural-ES, EPSL-R1, and EPSL, the number of solutions drawn is $N\lambda$, where $N$ and $\lambda$ refer to the number of sampled preferences and the number of solutions for each preference, respectively. Thus, the total samples across all generations are $N\lambda\times$ (total generations). As specified in Sec. C.2 of the supplementary document, $N=\lfloor5n/16\rfloor$ and $\lambda=2+⌊1.5\ln⁡{n}⌋$, and the total number of solutions is $4000n$ and $2000n$ for the bi- and tri-objective BBO-PSL instances, respectively.
>
> We plotted the convergence curves for all eight BBO-PSL instances in Fig. 15 of the supplementary document. To achieve the same IGD performance, Neural-ES always uses much fewer generations (and thus fewer samples) than EPSL and EPSL-R1. This highlights its sample efficiency, critical for black-box optimization.
>
> **Role of Adam Optimizer** Further efficiency gains arise from the use of Adam for training the set models in all three algorithms. Adam’s momentum-based gradient accumulation improves gradient estimation accuracy over time, which reduces the need for excessive sampling.
>
> ------
>
> > ### **W2**: Observations from Fig. 6
>
> **A2:** We sincerely apologize for any confusion caused by the insufficient explanation of Figure 6 due to space constraints. Figs. 6(a) to (c) depict the approximate Pareto sets in the first three dimensions of the search space (i.e., the $x_1-x_2-x_3$ subspace of the $x$-space). In these figures, the gray surface is the ground-truth PS, while the colored points indicate the sampled solutions by different algorithms during the testing time. The solutions produced by EPSL appear scattered and poorly aligned with preference vectors, while both EPSL-R1 and Neural-ES demonstrate better organization.
>
> Fig. 6(d) depicts the ground-truth Pareto front (i.e., surface with gray contours) and the algorithm approximations in the objective space (i.e., the $F$-space). For clarity, we excluded dominated solutions from visualization. This provides a more rigorous performance assessment because it directly evaluates solution quality in the objective space and reveals dominance relationships not apparent in decision space. While Neural-ES and EPSL-R1 appear similar in decision space (Figs. 6(b)-(c)), Neural-ES clearly outperforms in objective space, with most EPSL-R1 solutions being dominated by Neural-ES solutions.
>
> ------
>
> > ### **W3**: Results on the trajectory planning problem
>
> **A3:** We sincerely appreciate your valuable feedback regarding both the real-world application section and the presentation of results. Below, we provide both quantitative and additional experimental evidence to support our claims:
>
> | | Neural-ES | EPSL-R1 | EPSL |
> |:---------------------:|:---------:|:-------:|:-------:|
> | #Solutions dominated by Neural-ES | - | 194/300 | 299/300 |
> | HV (higher the better)      | **8.314** | 5.144 | 4.186 |
>
> We acknowledge that, according to Fig. 7, EPSL-R1 and Neural-ES possibly excel at different objectives. Nonetheless, because a non-dominated solution is always preferred over a dominated one, and HV usually directly reflects both convergence and diversity, we believe the results objectively demonstrate the efficacy of Neural-ES.
>
> __New experiments on UAV navigation__
> To further strengthen our claims, we conducted additional experiments on real-world tri-objective unmanned aerial vehicle (UAV) navigation problems with varying difficulty levels, with the following acquired results:
>
> | Difficulty | Neural-ES (HV) | EPSL (HV) | EPSL-R1 (HV) |
> |:----------:|:--------------:|:---------:|:------------:|
> | Easy | **3.397±0.279** | 2.471±0.356 | 1.577±0.149 |
> | Moderate | **3.443±0.452** | 2.399±0.372 | 1.520±0.267 |
> | Hard | **2.442±1.143** | 0.716±0.197 | 0.842±0.139 |
>
> The performance lead by Neural-ES across all difficulty levels validates its superiority in practical applications.
>
> __Availability of Raw Data__: We sincerely apologize for any confusion regarding the presentation format. Due to the space limit, we included representative visual results in the main text, while attaching in the supplementary document all raw quantitative data for producing Fig.4 (i.e., Table 2), as well as the convergence curves for all instances (i.e., Fig. 15).
>
> ------
>
> > ### **Q2**: Details about IGD and HV
>
> **A4:** Given a subset of the ground-truth Pareto front $\widehat{PF}$, and an approximate Pareto front $PF$ by an algorithm, the IGD value is given by
>
> $IGD(PF,\widehat{PF})=\frac{1}{|\widehat{PF}|}\sum\limits_{F^\star\in\widehat{PF}}\min\limits_{F\in PF}\|F-F^\star\|_2$.
>
> Intuitively, IGD measures the distance from each point $F^\star$ in the ideal $\widehat{PF}$, to the closest $F$ in $PF$. When imposing ideal properties on $\widehat{PF}$, the IGD is minimized only when the approximation set $PF$ spreads widely and evenly, or in other words, it converges to a good set. Otherwise, there will always be some $F^\star\in\widehat{PF}$ that remains distant from $PF$, resulting in a large IGD.
>
> The HV measures the volume of a specific region in the objective space, which is between a solution set and a reference point. It is given by:
>
> $HV_u(S)=\Lambda(\{y|\exists x\in S: F(x)\prec y\prec u\})$,
>
> where $S$ is a solution set, $u\in\mathbb{R}^M$ is a reference point in the $F$-space, $\Lambda$ denotes the Lebesgue measure, and $\prec$ denotes Pareto dominance.
> The volume always increases when the solution set approaches more closely to the ground truth PF or spreads more widely across the PF, as they both stretch the region.
>
> Both HV and IGD measure the convergence and diversity of a solution set to the ground-truth PF. IGD is favoured for its computational efficiency, while HV exhibits preferable properties such as submodularity and monotonicity [1,2]. For years, these two metrics have been widely adopted in the literature.
>
> ------
>
> > ### **Limitations**:
>
> **A5:** We appreciate your perspective on the scope of our limitation discussion. Constraints have consistently been a long-standing issue for both multiobjective optimization and evolution strategies [3,4], and we argue that it represents a fundamental technical barrier specific to our Neural-ES approach's current capabilities. Below, please see our detailed justification.
>
> __Manifold discontinuity__: Constrained MOPs often produce fragmented Pareto fronts [5]. This fundamentally breaks our current continuous mapping assumption between preference vectors and solutions.
>
> __Sampling breakdown__: The sampling of a random solution $x=(x_1^T,…,x_p^T )^T\sim\mathcal{N}(\varphi_\theta(x_{0:i-1}))$ becomes problematic when early segments $x_0$ to $x_{i-1}$ are frequently infeasible, causing cascading failures in generating valid $x_i,...,x_p$.
>
> ------
>
> > ### **References**
>
> [1] AP Guerreiro, CM Fonseca, L Paquete. "The hypervolume indicator: Computational problems and algorithms." ACM Computing Surveys (CSUR) 54.6 (2021): 1-42.
>
> [2] JG Falcón-Cardona, CAC Coello. "Indicator-based multi-objective evolutionary algorithms: A comprehensive survey." ACM Computing Surveys (CSUR) 53.2 (2020): 1-35.
>
> [3] D., Paul, and N. Hansen. "Augmented Lagrangian, penalty techniques and surrogate modeling for constrained optimization with CMA-ES." Proceedings of the Genetic and Evolutionary Computation Conference. 2021.
>
> [4] X. Chu,  M. Fei, and W. Gong. "Competitive multitasking for computational resource allocation in evolutionary constrained multi-objective optimization." IEEE Transactions on Evolutionary Computation (2024).
>
> [5] Z. Ma and Y. Wang. Evolutionary Constrained Multiobjective Optimization: Test Suite Construction and Performance Comparisons. IEEE Transactions on Evolutionary Computation, 23(6):972–986, December 2019.

---

> > ### Comment · Reviewer_37Ck · 2025-08-07
> >
> > I would like to thanks the authors for their answers and extra experiments conducted. If not already done, I am advising strongly to update the paper according to the discussions with the reviewers. I am satisfied with the answers and I am maintaining my evaluation.

---

> > > ### Author Response · Authors · 2025-08-08
> > > **Thank you**
> > >
> > > We greatly appreciate your supportive comment. Thank you for the positive score, and we are pleased that your concerns have been addressed. We will carefully incorporate all the discussed improvements into the updated manuscript.

---

> ### Author Response · Authors · 2025-08-06
> **We hope that our responses addressed your concerns**
>
> Thank you once again for your positive rating and for all your invaluable feedback. We hope our responses will adequately address your concerns. If you have any further questions, we would be more than happy to follow up.

---

### Official Review · Reviewer_zJXa · 2025-07-03

**Clarity:** 2
**Significance:** 3
**Originality:** 3
**Rating:** 4
**Confidence:** 3

**Summary:**

This paper proposes a NES-based optimization method to solve high-dimensional black-box multi-objective optimization problems (MOPs) via Pareto Set Learning (PSL). PSL maps user preferences to Pareto-optimal solutions using a set model (a neural network). Neural-ES partitions high-dimensional variables into segments, models dependencies via conditional Gaussian distributions parameterized by a neural network, and uses a novel natural gradient estimator for training. Notably, the method scales PSL to over 1000 dimensions.

**Questions:**

Please refer to 'weaknesses' in Strengths And Weaknesses.

**Ethical Concerns:**

["NO or VERY MINOR ethics concerns only"]

**Limitations:**

Yes.

**Quality:**

3

**Strengths And Weaknesses:**

Strengths:

1. The proposed method integrates ES with neural networks to capture complex dependencies in high-dimensional spaces, addressing a critical gap in black-box PSL.

2. The proposed method demonstrated scalability to 1024-Dim problems, which significantly beyond prior PSL methods.

3. The authors leverage natural gradient estimation and Fisher information for stable convergence, with proofs in supplementary materials.

Weaknesses

1. Neural-ES requires training a neural network and running ES evaluations. The wall-clock time vs. traditional ES/MOEAs is unreported, raising concerns about practicality.

2. Complexity claims of the proposed method focus on asymptotics but ignore constants (e.g., network inference costs).

3. BBO-PSL functions are synthetic. Real-world high-dimensional MOPs (e.g., neural architecture search) remain unvalidated.

4. The method assumes unconstrained MOPs, but real-world problems often include constraints. The limitation is acknowledged but unaddressed.

---

> ### Author Rebuttal · Authors · 2025-07-31
>
> We sincerely appreciate the reviewer's thoughtful and insightful feedback. Below, we provide detailed responses to the key concerns raised, with new experimental validations incorporated in the revised manuscript.
>
> ------
>
> > ### **W1:** Runtime assessment
>
> **A1:** We have conducted an extensive evaluation of computational efficiency across all compared algorithms using two 1024-dimensional instances (F1 and F5) from our BBO-PSL suite. The evaluation budget was the same as mentioned in Sec. C2 of the supplementary document, which is $4000n$ for F1, and $2000n$ for F5. The runtime results are as follows:
> | Algorithm | F1 ($n=1024$) | F5 ($n=1024$) |
> |:------------:|:------------------:|:------------------:|
> | LMO-CSO | 1019s (16.98min) | 1432s (23.87min) |
> | IM-MOEA/D | 1044s (17.40min) | 911s (15.18min) |
> | EPSL  | 486s (7.76min) | 308s (5.13min) |
> | EPSL-R1 | 731s (12.18min) | 443s (7.38min) |
> | Neural-ES | 1042s (17.37min) | 567s (9.45min) |
>
> We make the following observations. First, the PSL paradigm demonstrates remarkable computational efficiency. Both EPSL and EPSL-R1 achieve significantly faster runtimes (2-3×) compared to state-of-the-art MOEAs. This is because most MOEAs require advanced techniques to maintain population diversity and to deal with slow convergence, such as the RVEA-based environmental selection in LMO-CSO and the Gaussian process modeling in IM-MOEA/D. Most, if not all, of these techniques come with high overhead and scale poorly with increasing population sizes. In contrast, the PSL paradigm does not rely on any of these techniques. Despite employing neural networks, the PSL algorithms turn out to run surprisingly faster, or at least equally fast, when compared to traditional MOEAs.
>
> While Neural-ES shows marginally higher runtime compared to the other two PSL algorithms, due to its more sophisticated distribution modeling, it achieves dramatically faster convergence in terms of generations required. As shown in Fig. 15 of the supplementary document, Neural-ES needs only one-fifth to one-tenth the number of generations to reach the same IGD performance level as EPSL and EPSL-R1. This faster convergence likely offsets its marginally higher computational cost per generation, making it advantageous for practical applications.
>
> ------
>
> > ### **W2:** Complexity analyses
>
> **A2:** We analyzed the overall computational complexity, including the network inference costs, in Sec. B-7 of the supplementary document. In brief, Neural-ES achieves $\mathcal{O}(n)$ space complexity and $\mathcal{O}(n)$ time complexity when ignoring the network inference cost, as well as $\mathcal{O}(n^2)$ time complexity when taking into account the cost of the network. This complexity is identical to EPSL and EPSL-R1. This matches the theoretical lower bound for neural network-based gradient estimation, where the chain rule operation $\frac{\partial}{\partial\theta}\left(\frac{\partial\mathcal{J}}{\partial\varphi}\right)$ inherently requires $\mathcal{O}(n^2)$ computations when the distribution parameters $\varphi$ and the network parameters $\theta$ both scale linearly with dimension $n$. In light of the superior performance of Neural-ES, we believe the benefits justify the costs.
>
> ------
>
> > ### **W3:** Real-world problems
>
> **A3:** In Section 5.3, we presented an experiment on a real-world trajectory planning problem. However, in response to the valuable suggestion to validate our method on more practical problems, we have supplemented our work with new experiments focusing on multi-objective unmanned aerial vehicle (MO-UAV) navigation, which is featured in the Meta-BBO-v2 library [1,2]. This problem consists of three competing objectives, which are to be minimized:
>
> * the distance from the UAV's final location to the destination, penalized by the trajectory’s non-smoothness;
> * the risk of collision with threats, such as ground-to-air missiles;
> * an altitude penalty to maintain safe and cost-efficient flight.
>
> These objectives are thoroughly elaborated in [3]. By varying the control parameters of the terrain, we have developed three easy, moderate, and hard instances, which require a UAV trajectory of 10, 20, and 40 waypoints, respectively. Because each point comprises x, y, and z coordinates, the overall problem dimensions are 30, 60, and 120, respectively. MO-UAV serves as a typical real-world scenario where complex dimensional dependencies arise from temporal correlations between waypoints, which makes the navigation highly challenging.
>
> After 21 independent runs, we report the brief results in terms of the average HV values (higher is better) and their standard deviations as follows:
> | Instance | Dimensions | Neural-ES (HV) | EPSL (HV) | EPSL-R1 (HV) |
> |:---------:|:----------:|:--------------:|:------------:|:-------------:|
> | Easy | 30 | **3.397±0.279**  | 2.471±0.356 | 1.577±0.149 |
> | Moderate | 60 | **3.443±0.452**  | 2.399±0.372 | 1.520±0.267 |
> | Hard | 120 | **2.442±1.143**  | 0.716±0.197 | 0.842±0.139 |
>
> The significantly higher HV values across all difficulty levels confirm Neural-ES's effectiveness in practical multi-objective optimization scenarios.
>
> ------
>
> > ### **W4:** Limitations
>
> **A4:** We fully acknowledge the importance of constraint handling in multi-objective optimization, as rightly emphasized by the reviewer. Constraints are indeed a fundamental challenge in real-world applications. However, constraints pose particularly unique difficulties for Neural-ES, which stem from two core algorithmic characteristics:
> * __Manifold discontinuity__: Constrained MOPs often produce fragmented Pareto fronts with isolated regions. This disrupts Neural-ES’s key assumption of a full mapping from preference vectors to Pareto optimal solutions.
> * __Sampling breakdown__: The sampling in Neural-ES, where a segment $x_i$ depends on preceding $x_0,...,x_{i-1}$ becomes unrecoverable if early segments frequently violate constraints; besides, gradient-based training also fails when constraints block feasible ascent directions.
>
> While we acknowledge this critical gap, addressing it thoroughly would require further algorithmic changes, which may be too complex to cover in detail within the scope of this brief paper. We genuinely appreciate this critique and will prioritize it in subsequent research.
>
> ------
>
> > ### **References**
>
> [1] Z. Ma, et al. "Metabox: A benchmark platform for meta-black-box optimization with reinforcement learning." NeurIPS 36 (2023): 10775-10795.
>
> [2] Z. Ma, et al. "MetaBox-v2: A Unified Benchmark Platform for Meta-Black-Box Optimization." arXiv preprint arXiv:2505.17745 (2025).
>
> [3] MA Shehadeh, J Kůdela. "Benchmarking global optimization techniques for unmanned aerial vehicle path planning." Expert Systems with Applications (2025): 128645.

---

> > ### Comment · Reviewer_zJXa · 2025-08-03
> > **Thanks**
> >
> > Thanks for the detailed rebuttal, I'd like to maintain my original rating of this paper.

---

> ### Author Response · Authors · 2025-08-04
> **Thank you**
>
> Thank you for your response and support. We will continually improve both the main text and the supplementary document.

---

### Note · Authors · 2025-08-13

Dear AC,

We would like to express our gratitude to you for coordinating the discussion process, and to all the reviewers for their valuable feedback and engagement with our work. We are pleased to note that the reviewers found our proposed Neural-ES **straightforward and technically sound**. They all kindly **gave us positive ratings initially and continued to uphold those scores after the rebuttal**. We have made the following careful revisions according to the reviewers' suggestions, which have mostly addressed their concerns and will be added into the updated manuscript:

* __Runtime analyses__: Reviewers @zJXa, @37Ck, and @W93V raised concerns about the runtime of the proposed Neural-ES to ensure convergence (refer to W1, W1, and W3 by @zJXa, @37Ck, and @W93V). In response, we have measured the wall-clock time of the five compared algorithms. The results show that Neural-ES achieves competitive or even lower time costs against traditional MOEAs. Considering that Neural-ES allows for zero-shot inference after training, we can conclude that it is practically efficient.

* __More real-world applications__: Reviewers @zJXa, @37Ck, and @evfh expressed concerns about Neural-ES' performance in practical applications (refer to W3 by the reviewers). In response, we have provided a more comprehensive assessment using real-world tri-objective UAV path planning problems with varying dimensions and difficulty levels. The results indicate that Neural-ES significantly outperforms other algorithms, mainly due to its ability to decouple complex dimensional dependencies, such as those between waypoints in a trajectory.

* __Theorectical aspects__: We have engaged in extensive discussions with reviewer @evfh regarding the theoretical convergence of the broader class of ES and MOO algorithms (see W1 and W2 by @evfh and our responses). We would like to express our sincere gratitude to @evfh for the suggustions.

* __Clarifications__: Given the technical depth of Neural-ES and the space limit of the main text, we recognize that certain aspects may have initially been less transparently conveyed to the reviewers. In our response, we have carefully addressed each point of concern (e.g., W1, W2, Q1 and Q2 by W93V), and have provided references to specific sections or appendices where relevant.

We are fully prepared to present the strongest version of our work, shaped by the reviewers’ valuable feedback.

Kind regards,

Authors of submission 28972

---

### Decision · Program_Chairs · 2025-09-17

**Decision:**

Accept (poster)

**Comment:**

This paper proposes a neural evolution strategy for high-dimensional black-box multi-objective optimization problems via Pareto set learning (PSL). The Gaussian distributions modeled by a neural network are used for solution generation. The neural network is updated using the estimated gradient by leveraging the natural gradient of the Gaussian distribution. The proposed Neural-ES is evaluated using a benchmark suite for black-box PSL. The experimental result shows the efficiency of Neural-ES, demonstrating its capability to effectively learn the Pareto front in several multi-objective scenarios.

This paper integrates evolution strategies with neural network-based PSL, which is an interesting direction. The experimental results verify the effectiveness of the proposed method.

The authors provided extra experiments in the rebuttal and adequately answered the reviewers' concerns. Based on the reviewers' comments and discussion, the contribution of this paper meets the standards of acceptance.

I would encourage the authors to include additional explanations and experiments discussed in the rebuttal phase in the revised paper. I also think that it is important to justify the reformulation of the original problem using Eq. (4) as pointed out by the reviewer.

In addition, as a minor comment from AC, I could not find the setting of the learning rate $\eta$ for the proposed method. It would be better to clarify it and analyze the sensitivity of the learning rate setting. Also, in the rebuttal, the authors mentioned that the proposed method used the Adam optimizer, but I could not find its description in the main paper. It would be better to clarify it in the revised paper.